# Effects of Dietary Gamma-Aminobutyric Acid (GABA) Inclusion on Acute Temperature Stress Responses in Juvenile Olive Flounder (*Paralichthys olivaceus*)

**DOI:** 10.3390/ani15060809

**Published:** 2025-03-12

**Authors:** Abayomi Oladimeji Ogun, Haham Kim, Sooa Yoon, Suhyun Lee, Hyuncheol Jeon, Deni Aulia, Junhyeok Hur, Seunghyung Lee

**Affiliations:** 1Major of Aquaculture and Applied Life Sciences, Division of Fisheries Life Sciences, Pukyong National University, Busan 48513, Republic of Korea; abayomi.ogun@yahoo.com (A.O.O.); haham7@naver.com (H.K.); dbshelena@naver.com (S.Y.); su8842@naver.com (S.L.); conjp@naver.com (H.J.); damursalin@gmail.com (D.A.); junhyeok1999@naver.com (J.H.); 2Feeds and Foods Nutrition Research Center, Pukyong National University, Busan 48547, Republic of Korea

**Keywords:** aquaculture, climate change, stress relief, global warming, functional additive

## Abstract

The rapid warming of South Korean waters due to climate change presents a significant threat to marine aquaculture. As temperature stress becomes an increasing concern, nutritional strategies play a vital role in mitigating its impact on farmed species. While dietary gamma-aminobutyric acid (GABA) has been shown to mitigate stress in various species, research on its effects in mitigating temperature stress in olive flounder, a key aquaculture species in the Republic of Korea, remains limited. This study aims to assess the effects of dietary GABA inclusion on acute temperature stress in olive flounder (*Paralichthys olivaceus*). Therefore, we conducted analyses at various biological levels of organization after raising olive flounder juveniles on diets supplemented with different levels of GABA (control (GABA74), 174 ppm of GABA (GABA174), 275 ppm of GABA (GABA275), 396 ppm of GABA (GABA396), 476 ppm of GABA (GABA476), and 516 ppm of GABA (GABA516)) for 8 weeks. Although our observations suggest that GABA alone may not effectively mitigate acute temperature stress in this species, we cannot entirely rule out its efficacy, as several other factors may converge to influence the effect of GABA. Understanding these factors will be crucial in guiding future research on GABA in various aquaculture species, particularly those that may appear unresponsive to GABA inclusion under certain stress conditions.

## 1. Introduction

The olive flounder (*Paralichthys olivaceus*) is a commercially important species in East Asian aquaculture, particularly along Korea’s southern coast, where annual production exceeded 39,900 metric tonnes in 2023 [1]. However, the sustainability of this species is threatened by rising temperatures due to climate change. In the last 55 years (1968–2023), sea surface temperatures (SSTs) and ocean heat content in Korean waters have risen at approximately 2.6 times the global average, significantly impacting marine ecosystems and fish farming operations (Figure 1) [2,3,4,5]. Future projections by Kim et al. [6] indicate that SSTs in Korea could increase by an additional 1 to 4 °C by 2100, with the East Sea expected to experience the most substantial rise. Moreover, the National Institute of Fisheries Science in the Republic of Korea has been monitoring fluctuations in seawater temperatures, predicting an increase in the number of high-temperature days across the East, West, and South Seas and along the coast of Jeju Island, where olive flounder farming is prevalent [2,6].

Consequently, olive flounder farms are particularly vulnerable to these temperature changes due to their heavy reliance on direct seawater for daily aquaculture operations. This reliance makes it challenging to maintain optimal farming conditions [7]. Fish raised in intensive aquaculture systems are often subjected to stressors, such as elevated water temperatures, that exceed their optimum rearing limits [8,9], negatively impacting their health, growth, immune function, and reproductive success and posing a considerable threat to the sustainability of olive flounder farming. Excessive exposure to increased temperatures can also induce immune suppression, making the fish more susceptible to disease outbreaks [10]. Moreover, prolonged exposure to environmental stressors redirects the physiological energy typically dedicated to growth and metabolism, utilizing it to cope with stress [11], resulting in stunted growth, poor feed conversion, and ultimately, reduced farm profitability.

As temperature stress is difficult for olive flounder to avoid, it is becoming increasingly challenging to raise this species within its optimal conditions, which range from 20 to 25 °C [12]. During acute deviations from the optimal temperature range for olive flounder (e.g., sudden increases up to 30 °C [13,14]), both primary and secondary stress responses are triggered, depending on the duration and intensity of the exposure. The primary response usually occurs within minutes to an hour and involves the activation of the hypothalamic–pituitary–interrenal (HPI) axis, leading to cortisol release [15]. This release subsequently triggers secondary responses (occurring from hours to a few days after exposure) [16,17], including increases in plasma glucose, hematocrit, lactate, heart rate, gill blood flow, and metabolic rate, along with decreases in plasma chloride, sodium, potassium, liver glycogen, and muscle protein [18,19]. Upon prolonged exposure, tertiary responses (an outcome of the inability of the fish to regain homeostasis) occur, characterized by reduced growth, decreased metabolic scope, lowered disease resistance, impaired reproduction, and altered behavior and survival [20,21,22,23,24].

Dietary inclusions have been reported to improve the health of the fish and their resilience to stress [9,25,26,27]. One such inclusion is gamma-aminobutyric acid (GABA), a non-proteinogenic amino acid that serves as the primary inhibitory neurotransmitter in vertebrate brains [28,29]. GABA’s stress-reducing and sleep-enhancing effects are well established in humans [30]. As a key inhibitory neurotransmitter, GABA has been shown to regulate stress responses in various fish species by modulating the activation of the HPI axis and reducing cortisol release [7]. By alleviating the physiological burden of stress, GABA can help maintain immune function, improve growth, and enhance reproductive success. Additionally, GABA’s antioxidant properties can help combat oxidative stress, which is often exacerbated by environmental stressors such as temperature fluctuations. By enhancing antioxidant defenses, GABA can protect olive flounder from cellular damage, ensuring their health and resilience under challenging conditions. Furthermore, GABA’s ability to modulate the gut–brain axis may improve nutrient absorption and feed efficiency, leading to better growth outcomes.

However, recent studies suggest that its efficacy in aquaculture species may not be universal and could depend on factors such as species, duration of exposure, type of stress, and GABA purity. For example, in *Litopenaeus vannamei*, survival rates remained unchanged until 11 days of exposure, at which point GABA significantly improved survival in shrimp exposed to ammonia stress [31]. By contrast, in olive flounder (*Paralichthys olivaceus*) challenged with *Streptococcus iniae*, survival was significantly affected at 12 h postexposure [29]. Similarly, in *Oreochromis niloticus*, antioxidant gene expression was upregulated only after 96 h of ammonia exposure following a 56-day trial, indicating a positive response to GABA inclusion [28]. Additionally, Wang et al. [32] reported that in *Micropterus salmoides*, liver function enzymes, immune response, and antioxidant parameters varied according to the duration of GABA inclusion when the fish were exposed to ammonia stress, further highlighting the mitigatory role of GABA in this species when under ammonia stress.

These variations in response may partly stem from the difficulty of accurately replicating real-world environmental changes, including temperature fluctuations. This challenge is akin to the problems faced in designing appropriate simulation parameter distributions that align well with real-world conditions, without requiring exact replication [33]. Consequently, several studies examining the relationship between dietary manipulation and acute temperature stress have been conducted under controlled laboratory conditions [34,35]. Although the temperature setups in these previous studies are not ecologically relevant, since the complex functioning of biological systems cannot yet be fully replicated in controlled experimental designs [36], their findings provide valuable insights into the role of dietary manipulation in thermal stress responses for specific fish species.

Therefore, the objective of the current study aims to assess the effects of dietary GABA inclusion on stress responses of olive flounder to acute temperature stress under controlled laboratory conditions. Analyses at various biological levels of organization (from whole organism to molecules) after raising olive flounder juveniles on diets added with graded levels of GABA were investigated. Our observations from this study will provide valuable insights into the physiological responses of olive flounder during temperature fluctuations and offer a clearer understanding of considerations necessary to utilize GABA as a dietary mitigant to enhance stress resilience in aquaculture species.

## 2. Materials and Methods

The current study was reviewed and approved by the Institutional Animal Care and Use Committee (IACUC) of Pukyong National University, Busan, Republic of Korea (protocol code: PKNUIACUC-2023-14). The principal investigator and research team were committed to ensuring the humane and ethical treatment of all animals involved in this study.

### 2.1. Experimental Diet Preparation

The detailed experimental design is illustrated in Figure 2, and the composition of the basal diet is outlined in Table 1. Based on prior research by Farris et al. [29], the optimal GABA concentration was determined to be 237 mg/kg for weight gain. Our experimental diet concentrations were deliberately set below, within, and above this optimal threshold to allow for a comprehensive evaluation of GABA’s potential effects on the stress response of the fish. Ultimately, six isonitrogenous (approximately 54% crude protein), isolipidic (approximately 12% crude lipid), and isocaloric (approximately 5139 kcal/kg of gross energy) experimental diets were prepared, achieving actual GABA concentrations as follows: the control (which did not contain additional GABA) at 70 ppm (GABA70), and the test diets, containing additional GABA at 174 ppm (GABA174), 275 ppm (GABA275), 396 ppm (GABA396), 476 ppm (GABA476), and 516 ppm (GABA516). These concentrations were confirmed through high-performance liquid chromatography (HPLC).

Before producing the experimental diets, a GABA premixture was prepared. This was achieved by mixing 190 g of alpha-cellulose with 10 g of GABA (99% purity, Sigma Aldrich, St. Louis, MO, USA) to achieve a final premixture concentration of 500 ppm. This premixture was then thoroughly blended for 72 h using a ball mill machine (Model PL-BM5L, Poong Lim Trading Co., Seoul, Republic of Korea).

The feed formulation process began by mixing all the dry ingredients for 15 min using an electric mixer (HYVM-1214, Hanyoung Food Machinery, Hanam-si, Republic of Korea). Fish oil was then gradually added to the mixture, followed by an additional 15 min of mixing. Next, tap water, which constituted 45% of the total ingredient volume, was incorporated, and the mixture was thoroughly blended for another 15 min. The resulting wet dough was passed through a pelletizing machine (SFD-GT, Shinsung Co., Siheung-si, Republic of Korea) fitted with a flat die featuring holes approximately 2 mm in diameter. The pelleted strands were manually fragmented into smaller pieces and then oven-dried using a dryer (KE-010, Dongwon Industries, Seoul, Republic of Korea) at 45 °C for 18 h to reduce the moisture content. This process resulted in a moisture content below 10% after cooling. Once cooled to room temperature, the experimental diets were packaged, labeled according to their respective compositions, and securely stored in a freezer at −20 °C until scheduled for use.

The proximate composition of the experimental diets (Table 1) was determined using the methodology established by the Association of Official Analytical Chemists [37]. The moisture content was assessed by drying the homogenized diet samples to a constant weight at 105 °C. The ash content was quantified by incinerating the samples at 550 °C in a muffle furnace for 3 h. Nitrogen content (N × 6.25) was assessed through the Kjeldahl method, which involves acid digestion using the 2300 auto analyzer (Foss Tecator AB, Höganäs, Sweden). The crude lipid content was determined by ether extraction using the Soxtec system 1046 (Foss Tecator AB, Höganäs, Sweden). The free amino acid profile of the diets was analyzed using the methodology described by Antoine et al. [38].

### 2.2. Experimental Fish and Feeding Trial

The juveniles, with an initial mean weight of 10.7 ± 0.6 g (mean ± SEM), were sourced from a local fish farm (Samboo farm, Boryeong-si, Republic of Korea) and acclimated to the experimental conditions at the Aquafeed Nutrition Laboratory of the College of Fisheries, Pukyong National University, Busan. During the initial phase of acclimation, they were fed a commercial diet (crude protein 53.2%, crude lipid 10.5%, and crude ash 15%; Woosung Feed Co., Daejeon, Republic of Korea) for one week. Following this, they underwent a progressive transition to the control diet (GABA70), beginning with 25% inclusion of the control diet for 2 days, 50% inclusion of the control diet for the next 2 days, and then 100% inclusion of the control diet for the remaining acclimation days. The acclimation phase concluded after the feeding response reached a level of vigor comparable to that observed during the initial stage of feeding with the commercial diet. The overall acclimation period lasted a total of three weeks.

A total of 360 juveniles with an initial weight averaging 12.97 ± 0.1 g were randomly distributed into 18 tanks (20 fish per tank). The six experimental diets were assigned randomly to the tanks in triplicate. An additional group of extra fish of the same size was kept in a separate tank of equal volume within the same rearing system and labeled “Extra”. This extra set was fed the control diet throughout the experimental period and was later used for a pre-exposure evaluation before the temperature stress exposure test.

Feeding occurred twice daily at 9:00 and 17:00 over 8 weeks, with a fixed feeding rate of 2–3% of the total body weight per day. Every two weeks, all fish in the tanks were weighed, and the feed amount was adjusted accordingly. After feeding, a 30 min interval was allowed before fecal materials were removed.

The tanks were connected to a semi-recirculatory system equipped with a protein skimmer (Model ORI-M, Dongyangeng Co., Paju-si, Republic of Korea) that removed organic compounds, proteins, and other impurities from the water. An electronic thermostat (DOV-887, Daeil Co., Busan, Republic of Korea) ensured a constant water temperature, which was monitored by a HOBO data logger (HOBO water temperature Pro v2 data logger-U22-001, Onset, Bourne, MA, USA) every 10 min throughout the trial. A flow rate of 2.5 L/min was maintained during the experimental period. The system was continuously replenished with fresh, UV-sterilized seawater after siphoning to compensate for water loss, and a complete water change was performed twice a week to maintain optimal conditions. The photoperiod remained consistent, with 12 h of light followed by 12 h of darkness, facilitated by an artificial lighting system.

Daily water quality assessments included measurements of dissolved oxygen (DO) and pH using a multiparameter meter (YSI Model 58, Yellow Springs, OH, USA). Nitrogenous wastes were monitored using a commercial kit, i.e., API Saltwater Master Test Kit (Petco, San Diego, CA, USA). The water temperature was kept constant using an electronic thermostat (DOV-887, Daeil Co., Busan, Republic of Korea) and maintained at 19.5 ± 0.1 °C. DO levels were recorded at 7.98 ± 0.18 mg/L, pH at 7.38 ± 0.02, ammonia at 0.84 ± 0.06 mg/L, nitrite at 0.82 ± 0.11 mg/L, and nitrate at 53.4 ± 2.3 mg/L throughout the experiment.

### 2.3. Growth Performance

At the end of the 8-week experiment, all fish (24 h postprandial) in each tank were weighed to calculate the growth performance indices, namely weight gain (WG), specific growth rate (SGR), feed efficiency (FE), feed conversion ratio (FCR), and survival rate (SR). The following equations were used for each index:
WG (%) = [(final weight (g) − initial weight (g))/initial weight (g)] × 100SGR (%) = [ln (final weight(g)) − ln (initial weight (g))]/days of feeding × 100FE (%) = [weight gain/total feed consumed (g)] × 100FCR = feed weight as dry (g)/wet weight gain (g)SR (%) = [(initial number of fish − number of dead fish)/initial number of fish] ×100


Three fish from each tank were euthanized with phenoxyethanol (500 ppm; Sigma Aldrich, St. Louis, MO, USA) and then stored in a freezer (−20 °C) until further analysis. The frozen fish were pooled, homogenized, and freeze-dried for whole-body proximate composition analysis using the methodology established by the AOAC [36].

### 2.4. Temperature Stress Exposure Test

#### 2.4.1. Lethal Temperature Exposure

Before conducting the main lethal temperature exposure test, we performed a pre-exposure test to evaluate the suitability of the target temperature. Eight extra fish, which were kept on the control diet throughout the experimental period, were exposed to the same conditions planned for the main test. No mortality was observed before reaching the target temperature of 31 °C during this pre-test, confirming that the chosen lethal temperature was appropriate for the main test.

Subsequently, to investigate the whole-organism responses of the test-diet-fed experimental fish over 8 weeks, eight 24 h postprandial fish that were subjected to lethal temperature exposure. The temperature was incrementally raised at a rate of 1 °C every 30 min, starting from the rearing temperature of 19.5 °C and eventually reaching the target temperature of 31 °C. This approach was similar to the procedure outlined by Lu et al. [39]. Fish survival was monitored and recorded hourly for a continuous 48 h period, at which point the lethal stress experiment concluded. All tanks were equipped with aeration systems to ensure that dissolved oxygen levels remained near saturation. During the exposure test, a data logger (HOBO Water Temperature Pro v2 data logger-U22-001, Bourne) was used to monitor the temperature throughout the experiment, recording data every 10 min. Dissolved oxygen (DO) levels were measured and recorded every hour using a multiparameter meter (YSI Model 58, Yellow Springs, OH, USA). All tanks were equipped with aeration systems and a centralized oxygenator to ensure that DO levels consistently remained at 7.84 ± 0.06 mg/L.

#### 2.4.2. Acute Temperature Exposure

A group of three test-diet-fed 24 h postprandial juveniles per tank was abruptly transferred into separate tanks filled with seawater at an elevated temperature of 29 °C. This elevated temperature exposure lasted for six hours. The selection of the temperature and duration of exposure was based on the outcomes of a preliminary test conducted with the same group of juveniles. Importantly, this preliminary test revealed no observable mortality within 24 h postexposure, indicating that the selected conditions did not result in ecological mortality and were therefore suitable for acclimation in our experiment. To investigate any stress (e.g., handling stress) other than temperature stress during the acute test, an additional set of three juveniles from each tank was transferred to a separate tank maintained at the rearing temperature of 19.5 ± 0.1 °C. This handling stress exposure was sustained for six hours, aligning with the duration of the acute stress exposure.

At the end of the exposure test, the fish were euthanized using 2-phenoxyethanol (500 ppm, Sigma Aldrich, St. Louis, MO, USA). Blood samples were then collected from the caudal vein using a 1 mL syringe pre-treated with dipotassium ethylenediaminetetraacetic acid (EDTA; Bylabs, Hanam-si, Republic of Korea) as an anticoagulant. The collected blood samples from each tank were pooled into a 3 mL siliconized vacuum tube treated with EDTA, divided into 1 mL microtubes, and immediately centrifuged at 11,000 rpm for 5 min. The resulting clear plasma was decanted into 1.5 mL microtubes and snap-frozen in liquid nitrogen until all samples were collected. The plasma-filled microtubes were stored at −84 °C until further analysis. Additionally, whole-brain and liver samples were extracted from each individual and labeled with aluminum foil according to their respective tanks. These tissues were also snap-frozen in liquid nitrogen until all samples were collected and subsequently stored at −84 °C until further analysis.

#### 2.4.3. Temperature Stress Tolerance Assessment

##### Free Amino Acid Analysis

A free amino acid analysis of the samples was conducted after the acute temperature exposure using the method described by Antoine et al. [38]. A total of 0.3 g of each tissue (liver and intestine) was homogenized using a hand-held homogenizer (Model D-130, Wiggens Co. Ltd., Beijing, China). After homogenization, 10 mL of HPLC-grade distilled water was added, and the mixture was vortexed for 1 min at 5000 rpm using a high-performance multifunction vortex mixer (Maxshake^TM^, Daihan Scientific, Wonju-si, Republic of Korea). This was followed by sonication for 20 min. The samples were then centrifuged at 4500 rpm for 10 min at 4 °C. A total of 0.7 mL of the resulting supernatant was placed into 1.5 mL microtubes, and 0.7 mL (ratio 1:1) of 7% sulfosalicylic acid was added. The mixture was kept in the dark at 4 °C overnight. Afterward, it was centrifuged at 4500 rpm for 10 min at 4 °C, and 1 mL of the supernatant was passed through a 0.2-micron filter and dispensed into HPLC vials for analysis. For the plasma samples, the same steps were followed, except for the homogenization step, which was skipped due to the liquid nature of the samples.

##### Plasma Metabolites and Biomarker Analyses

The plasma samples from the stress and non-stress groups (handling) collected during the acute exposure test were analyzed for plasma metabolites, including glucose (GLU; code 1050), total cholesterol (TCHO; code 1450), triglycerides (TG; code 1650), total protein (TP; code 1850), glutamic oxaloacetic transaminase (GOT; code 3150), and glutamic pyruvic transaminase (GPT; code 3250). The analysis was conducted using a dry biochemical automatic analyzer (Fuji DRI-CHEM nx500i, Fuji Photo Film, Tokyo, Japan), following the manufacturer’s instructions.

Enzyme-linked immunosorbent assay (ELISA) quantification kits (CUSABIO, Wuhan, China) were utilized, according to the manufacturer’s instructions, and a microplate reader (AMR-100, Allsheng, Hangzhou, China) set to an optical density of 450 nm facilitated the quantification process. Analyses included the assessment of antioxidant system indicators such as glutathione peroxidase (GPx; code CSB-E15930Fh) and superoxide dismutase (SOD; code CSB-E15929Fh). Additionally, immunological parameters were evaluated as stress-related indicators in the plasma samples obtained after the acute exposure test, including lysozyme (LZM; code CSB-E17296Fh), immunoglobulin M (IgM; code CSB-E12045Fh), and cortisol (CORT; code CSB-E08487Fh).

##### Molecular Response

An analysis was conducted on the brains and livers of the fish subjected to the acute temperature exposure test. The samples were collected from three individuals per tank and pooled within each tank. All 18 experimental tanks, corresponding to the six experimental groups (N = 3), were utilized in the analysis. From each pooled group (tank), less than 100 mg of tissue was sampled and homogenized in a microtube containing 1 mL of RiboEx^TM^ (GeneAll Biotechnology Co., Ltd., Seoul, Republic of Korea) using a homogenizer (Model D-130, Wiggens Co., Ltd., Beijing, China). Total RNA was extracted using a commercial kit (Hybrid-R^TM^, GeneAll, Seoul, Republic of Korea), following the manufacturer’s protocol. The concentration and purity of the RNA were checked using a NanoDrop ASP-2680 spectrophotometer (ACTgene, Piscataway, NJ, USA). It was ensured that the purified RNA was free of DNA and proteins, with an A260/A280 ratio of 2.21 ± 0.01 (mean ± SEM). Complementary DNA (cDNA) was synthesized using a commercial kit (PrimeScript^TM^ 1st Strand cDNA Synthesis Kit, Takara Bio, Shiga, Japan), following the manufacturer’s protocol. The obtained cDNAs were used as templates for quantitative real-time PCR (qRT-PCR) to assess mRNA expression levels of four selected stress-related genes: heat shock protein 60 kDa (*hsp60*), heat shock protein 70 kDa (*hsp70*), heat shock protein 90 kDa (*hsp90*), and warm temperature acclimation-related protein 65 kDa (*wap65*). Beta-actin (*β-actin*) was used as the internal control. The primer sequences for the genes are listed in Table 2. The reaction was performed using the StepOne Real-Time PCR system (Applied Biosystems, Waltham, MA, USA) using a TB^®^ Green Premix Ex Taq^TM^ II device (Tli RNaseH Plus, Takara Bio, Shiga). The qPCR program was set to 95 °C for 30 s, followed by 40 cycles of 95 °C for 5 s and 60 °C for 30 s. After the reaction, a melting curve analysis was conducted to verify the specificity of the products. For each selected gene, qPCR reactions were performed on three replicate samples. The mRNA levels of the tested genes were normalized to the corresponding *β-actin* value and analyzed using the 2^−△△Ct^ method [40]. Data analysis was performed using StepOne Software version 2.0 (Applied Biosystems, Waltham, MA, USA).

### 2.5. Statistical Analysis

Results of the growth response and whole-body proximate composition obtained following the 8-week feeding trial, as well as the free amino acid profile in the brain, intestine, and plasma acquired from the acute temperature exposure test, were subjected to a one-way analysis of variance (ANOVA) using SAS version 9.4 [45]. Before conducting the ANOVA, the data integrity was evaluated for the assumptions, including normality and homogeneity of variance, using the Shapiro–Wilk and Levene’s tests, respectively. The results of these tests indicated that the assumptions of normality and homogeneity were met. Statistical significance was determined at *p* < 0.05, and all analyses were performed at a 95% confidence level.

For the survival analysis, we used R version 4.4.1 [46] with the “survival” and “survminer” packages to perform Kaplan–Meier survival analysis. This method was chosen for its robustness in estimating survival curves and comparing survival rates among groups. The survfit function in the “survival” package was used to generate survival curves, and the log-rank test was applied to compare survival rates across groups. As with other analyses, significance was tested at *p* < 0.05 to ensure consistency.

The results of the plasma metabolites, biomarker analyses, and relative gene expression levels obtained from the acute temperature exposure test were subjected to a two-way ANOVA using SAS version 9.4 [45]. This analysis was conducted to evaluate the interactive effects between dietary treatment (different GABA levels in the diet) and temperature stress (non-stress and stress exposure). Type III sum of squares in ANOVA was employed to test all possible interactions and main effects. Statistical significance was determined at *p* < 0.05. When significant differences were detected, a post hoc test using Tukey’s HSD test was performed for pairwise comparisons to assess differences between treatment groups.

Additionally, orthogonal polynomial regression analyses were performed in SAS version 9.4 [45] using the PROC REG procedure. These analyses evaluated whether the measured responses to the graded GABA levels exhibited linear or quadratic relationships. All statistical methods and criteria, including tests for normality and homogeneity of variance, were implemented in line with standard protocols to ensure the reliability and reproducibility of the findings.

## 3. Results

### 3.1. Effects of GABA on Growth Performance and Body Composition

Table 3 illustrates the effects of dietary GABA inclusion on the growth performance of juvenile olive flounder throughout the 8-week experimental period. Overall, survival rates were high, exceeding 98% across all groups, with no statistically significant differences observed among them (*p* > 0.05). GABA inclusion at a 516 ppm inclusion level resulted in the greatest improvements in final body weight, weight gain, and specific growth rate. By contrast, the 174 ppm inclusion exhibited the highest feed efficiency. However, despite these observed trends, no statistically significant differences were found across all parameters related to growth and nutrient utilization. Similarly, neither the linear nor quadratic trends were statistically significant.

The results of the whole-body composition analysis of juvenile olive flounder fed the experimental diets for 8 weeks (Table 3, bottom) showed that dietary GABA inclusion at all levels did not result in significant differences in whole-body moisture, crude protein, crude lipid, or crude ash contents at the end of the experimental period.

### 3.2. Temperature Stress Tolerance Assessment

#### 3.2.1. Lethal Temperature Exposure

The survival rate of olive flounder exposed to lethal temperature stress for 48 h after the 8-week feeding trial is depicted in the Kaplan–Meier survivorship curve (Figure 3). The first occurrence of mortality in all the diet groups occurred after the first 8 hours of exposure, except in the GABA476 group, in which the first occurrence occurred within the first 8 h. However, the fish survival rates among all the experimental groups were not significantly different at the end of the 48 h exposure period. A clear differentiation between groups emerged after approximately 20 h of exposure. Higher GABA concentrations, such as GABA516, led to prolonged survival, while intermediate concentrations, such as GABA174 and GABA396, led to the highest declines in survival, although there was no significant difference in the survival rate average of 28.5 ± 4.6% (*p* > 0.05), as determined via a log-rank test.

#### 3.2.2. Effects of Temperature on Amino Acid Profile

Table 4a–c display the free amino acid profile in the whole brains, intestines, and plasma of juvenile olive flounder following the acute temperature exposure test. The results indicate that varying levels of GABA did not lead to any significant differences in these tissues in response to the stress (*p* > 0.05).

#### 3.2.3. Effects of Temperature and Dietary GABA on Plasma Metabolites and Biomarkers

The results of varying levels of GABA inclusion on juvenile olive flounders’ plasma metabolites in response to acute temperature exposure are presented in Table 5. At 19.5 °C, glutamate oxaloacetate transaminase (GOT) ranged from 14.3 ± 0.9 U/L in the GABA396 group to 19.7 ± 2.6 U/L in the GABA516 group. Glutamate pyruvate transaminase (GPT) ranged from 13.0 ± 0.6 U/L in the GABA396 group to 16.7 ± 2.3 U/L in the GABA275 group. 

Glucose (GLU) levels ranged from 12.0 ± 1.7 mg/dL in the control group to 15.0 ± 2.1 mg/dL in the GABA275 group. Triglycerides (TGs) ranged from 194 ± 22.8 mg/dL in the GABA174 group to 273 ± 50 mg/dL in the GABA275 group. Total protein (TP) ranged from 2.83 ± 0.10 mg/dL in the GABA275 group to 3.10 ± 0.10 mg/dL in the GABA396 group, and total cholesterol (TCHO) ranged from 137 ± 3 g/L in the GABA396 group to 171 ± 13 g/L in the GABA516 group, indicating no significant difference in these values. However, the main effect of temperature was observed, significantly elevating the values for GOT (46.3 ± 6.2 U/L), GPT (30.2 ± 2.1 U/L), GLU (68.7 ± 7.7 mg/dL), and TP (3.21 ± 0.1 mg/dL). By contrast, the values for TG and TCHO remained relatively stable, showing levels similar to those observed at 19.5 °C.

**Table 4 animals-15-00809-t004:** (**a**) Free amino acid profile in the brains of juvenile olive flounder exposed to the acute temperature stress. (**b**) Free amino acid profile in the intestines of juvenile olive flounder exposed to the acute temperature stress. (**c**) Free amino acid profile in the plasma of juvenile olive flounder exposed to the acute temperature stress ^1^.

(a)									
Free Amino Acids (ppm)	Diets						Pr > F		
	GABA70	GABA174	GABA275	GABA396	GABA476	GABA516	ANOVA	Linear	Quadratic
Phosphoserine	14.4 ± 1.7 ^ns^	20.7 ± 1.9	14.4 ± 3.2	16.7 ± 0.9	18.0 ± 1.3	19.7 ± 0.6	0.1348	0.5838	0.8267
Taurine	259 ± 18 ^ns^	295 ± 10	241 ± 36	275 ± 5	290 ± 1	291 ± 5	0.2589	0.4582	0.4938
Phosphoethanol amine	18.1 ± 5.9 ^ns^	24.6 ± 2.8	15.3 ± 5.7	18.3 ± 1.3	24.4 ± 1.0	24.0 ± 3.6	0.4193	0.621	0.4517
Aspartic acid	31.8 ± 7.1 ^ns^	40.0 ± 3.8	26.2 ± 8.7	33.3 ± 1.4	39.6 ± 1.5	39.1 ± 3.3	0.3691	0.5873	0.3881
Threonine	15.7 ± 4.6 ^ns^	15.7 ± 0.7	14.8 ± 6.0	17.6 ± 1.2	19.4 ± 4.4	19.0 ± 0.8	0.9142	0.4327	0.5923
Serine	35.1 ± 8.8 ^ns^	34.8 ± 1.2	32.4 ± 13.3	38.6 ± 3.1	40.5 ± 6.5	41.4 ± 1.3	0.9337	0.5341	0.6336
Glutamic acid	135 ± 32 ^ns^	191 ± 26	110 ± 39	147 ± 15	181 ± 18	174 ± 18	0.2783	0.5800	0.4551
Proline	6.22 ± 3.11 ^ns^	5.61 ± 0.81	7.55 ± 3.80	8.38 ± 1.81	8.53 ± 3.18	7.93 ± 0.81	0.9471	0.3768	0.9667
Glycine	19.3 ± 4.3 ^ns^	23.1 ± 1.8	16.3 ± 6.5	21.3 ± 0.4	22.5 ± 2.6	24.2 ± 0.2	0.6256	0.6776	0.6125
Alanine	39.4 ± 12.8 ^ns^	38.3 ± 1.3	36.6 ± 15.1	44.9 ± 5.5	54.1 ± 11.0	52.2 ± 3.7	0.7029	0.2602	0.4127
Valine	8.65 ± 3.06 ^ns^	6.72 ± 1.42	9.44 ± 4.02	10.24 ± 1.69	10.61 ± 3.78	10.01 ± 1.03	0.922	0.4105	0.7981
Cystine	0.00 ± 0.00 ^ns^	0.31 ± 0.31	0.22 ± 0.22	0.59 ± 0.17	0.33 ± 0.20	0.34 ± 0.18	0.5028	0.1584	0.3882
Methionine	13.6 ± 3.6 ^ns^	14.3 ± 0.5	12.6 ± 5.1	14.3 ± 0.7	12.1 ± 5.9	15.6 ± 0.5	0.9822	0.8067	0.8619
Isoleucine	6.74 ± 2.49 ^ns^	5.61 ± 0.78	7.29 ± 3.08	7.94 ± 0.89	8.21 ± 2.82	7.85 ± 0.66	0.9464	0.4345	0.8221
Leucine	16.7 ± 5.8 ^ns^	14.4 ± 2.3	17.0 ± 7.2	18.9 ± 2.1	19.4 ± 5.6	19.2 ± 1.7	0.9652	0.5098	0.7802
Tyrosine	8.05 ± 2.82 ^ns^	6.15 ± 1.08	8.26 ± 3.50	9.14 ± 1.31	8.45 ± 2.77	8.16 ± 1.01	0.9613	0.6136	0.8909
Phenylalanine	10.6 ± 9.59 ^ns^	9.59 ± 1.0	10.25 ± 4.1	12.03 ± 1.0	12.54 ± 3.01	11.8 ± 0.7	0.945	0.4247	0.6652
b-amino isobutyric acid	0.00 ± 0.00 ^ns^	0.00 ± 0.00	0.00 ± 0.00	0.09 ± 0.90	1.50 ± 1.50	0.00 ± 0.00	0.5565	0.1099	0.4486
γ-aminobutyric acid	62.5 ± 17.2 ^ns^	80.7 ± 0.5	62.3 ± 22.3	71.6 ± 3.6	84.3 ± 3.9	82.9 ± 2.9	0.6124	0.3713	0.709
Histidine	74.6 ± 18.8 ^ns^	57.8 ± 4.4	79.0 ± 32.9	85.7 ± 10.7	89.9 ± 22.0	83.8 ± 2.6	0.8528	0.3373	0.6994
Ammonia	0.00 ± 0.00 ^ns^	0.15 ± 0.15	0.00 ± 0.00	0.00 ± 0.00	0.00 ± 0.00	0.12 ± 0.12	0.5682	0.5608	0.6223
Arginine	27.7 ± 10.3 ^ns^	20.6 ± 0.9	23.9 ± 9.6	28.6 ± 3.2	25.6 ± 3.2	26.0 ± 2.3	0.9461	0.8451	0.6831
(**b**)									
**Free Amino Acids (ppm)**	**Diets**						**Pr > F**		
	**Control**	**GABA174**	**GABA275**	**GABA396**	**GABA476**	**GABA516**	**ANOVA**	**Linear**	**Quadratic**
Phosphoserine	4.46 ± 0.32 ^ns^	4.72 ± 0.82	5.17 ± 0.75	4.58 ± 0.67	4.75 ± 0.77	5.51 ± 1.37	0.9456	0.8683	0.7024
Taurine	68.5 ± 7.7 ^ns^	75.7 ± 9.0	83.2 ± 10.0	83.6 ± 5.5	77.6 ± 11.0	89.0 ± 13.0	0.7358	0.4115	0.3758
Aspartic acid	19.8 ± 2.8 ^ns^	19.4 ± 2.6	22.3 ± 4.2	22.9 ± 1.0	21.2 ± 3.2	27.0 ± 4.4	0.6156	0.5474	0.6948
Threonine	23.7 ± 3.5 ^ns^	24.2 ± 3.8	27.3 ± 5.1	28.5 ± 1.6	26.3 ± 3.9	34.2 ± 6.5	0.5831	0.5084	0.6584
Serine	29.5 ± 4.3 ^ns^	30.6 ± 4.2	34.8 ± 6.4	35.4 ± 1.7	32.0 ± 4.3	41.4 ± 7.6	0.6304	0.5592	0.5243
Asparagine	45.7 ± 4.4 ^ns^	51.1 ± 8.3	55.0 ± 8.8	58.6 ± 3.3	49.8 ± 4.0	70.4 ± 15.4	0.4295	0.5694	0.3841
Glutamic acid	53.3 ± 6.7 ^ns^	53.9 ± 8.4	61.1 ± 11.0	64.7 ± 3.3	56.8 ± 7.3	76.6 ± 14.7	0.5177	0.5559	0.5629
a-amino adipic acid	2.89 ± 0.21 ^ns^	3.18 ± 0.53	3.29 ± 0.47	3.50 ± 0.21	3.00 ± 0.33	4.19 ± 0.72	0.4145	0.7164	0.387
Proline	15.0 ± 1.7 ^ns^	15.9 ± 2.0	17.4 ± 2.7	17.8 ± 0.9	11.7 ± 6.1	20.6 ± 3.4	0.5361	0.6483	0.2405
Glycine	17.0 ± 2.4 ^ns^	17.6 ± 2.2	20.2 ± 3.6	21.0 ± 0.8	18.2 ± 2.1	24.1 ± 4.6	0.5404	0.5256	0.4342
Alanine	33.2 ± 4.9 ^ns^	34.2 ± 4.4	39.2 ± 7.5	40.8 ± 1.6	36.0 ± 4.5	46.8 ± 8.6	0.5858	0.5123	0.4995
Valine	20.0 ± 3.3 ^ns^	20.4 ± 2.9	23.1 ± 4.4	23.8 ± 2.2	23.2 ± 3.4	26.9 ± 4.1	0.7516	0.3898	0.7579
Cystine	2.76 ± 0.33 ^ns^	3.60 ± 0.88	3.15 ± 0.84	4.95 ± 0.60	3.61 ± 0.57	6.39 ± 2.44	0.3111	0.4246	0.639
Methionine	21.0 ± 2.6 ^ns^	22.3 ± 3.4	24.7 ± 4.8	26.1 ± 1.7	23.5 ± 2.9	31.0 ± 6.0	0.5333	0.4740	0.5485
Isoleucine	15.4 ± 2.3 ^ns^	15.8 ± 2.7	18.0 ± 4.0	18.4 ± 1.8	17.0 ± 1.9	22.8 ± 5.1	0.6319	0.5784	0.6687
Leucine	49.9 ± 7.1 ^ns^	52.4 ± 9.1	58.4 ± 11.4	61.6 ± 4.3	54.9 ± 7.0	75.5 ± 16.0	0.5308	0.5497	0.5761
Tyrosine	24.5 ± 4.4 ^ns^	25.0 ± 4.7	29.1 ± 6.1	30.08 ± 3.1	25.2 ± 2.8	35.6 ± 7.5	0.6145	0.6870	0.4813
Phenylalanine	22.2 ± 2.4 ^ns^	23.3 ± 4.2	25.3 ± 5.5	27.4 ± 0.9	24.5 ± b2.6	35.6 ± 7.7	0.3722	0.5540	0.6459
γ-aminobutyric acid	7.74 ± 1.20 ^ns^	8.77 ± 1.13	10.7 ± 1.6	10.4 ± 1.8	9.62 ± 1.63	12.0 ± 2.2	0.5407	0.3160	0.3627
Histidine	9.92 ± 1.46 ^ns^	10.4 ± 1.7	11.6 ± 2.2	12.1 ± 0.7	11.0 ± 1.2	14.3 ± 2.8	0.599	0.5131	0.5810
Lysine	19.1 ± 3.0 ^ns^	19.9 ± 2.8	23.1 ± 5.1	25.4 ± 0.7	22.0 ± 2.4	28.3 ± 5.9	0.5256	0.3471	0.5156
Ammonia	9.79 ± 1.00 ^ns^	8.30 ± 0.85	10.8 ± 1.1	11.6 ± 0.4	10.2 ± 1.8	11.2 ± 1.7	0.5121	0.3063	0.7737
Arginine	24.2 ± 2.3 ^ns^	31.8 ± 6.0	34.8 ± 7.2	37.1 ± 0.9	28.5 ± 3.2	42.3 ± 8.1	0.2769	0.4248	0.1222
(**c**)									
**Free Amino Acids (ppm)**	**Diets**						**Pr > F**		
	**Control**	**GABA174**	**GABA275**	**GABA396**	**GABA476**	**GABA516**	**ANOVA**	**Linear**	**Quadratic**
Phosphoserine	10.3 ± 2.2 ^ns^	11.3 ± 1.7	10.5 ± 1.8	12.5 ± 1.3	13.6 ± 1.0	14.2 ± 0.5	0.3834	0.1252	0.6211
Taurine	5.11 ± 0.89 ^ns^	5.86 ± 1.86	5.39 ± 1.30	5.92 ± 0.31	6.06 ± 0.33	7.22 ± 0.72	0.7869	0.5659	0.9542
Aspartic acid	1.25 ± 0.31 ^ns^	1.51 ± 0.54	1.32 ± 0.31	1.34 ± 0.33	1.51 ± 0.20	1.90 ± 0.09	0.7534	0.7427	0.9900
Threonine	5.74 ± 1.80 ^ns^	6.80 ± 1.09	5.90 ± 0.83	6.37 ± 1.30	7.62 ± 1.17	8.49 ± 0.77	0.5866	0.4008	0.7069
Serine	7.30 ± 2.05 ^ns^	8.76 ± 1.48	7.87 ± 1.40	8.91 ± 1.58	10.02 ± 1.31	10.5 ± 0.7	0.6483	0.2537	0.8289
Asparagine	11.6 ± 3.9 ^ns^	13.5 ± 0.70	11.9 ± 1.8	13.7 ± 2.4	16.3 ± 3.3	16.1 ± 1.9	0.6932	0.2588	0.6321
Glutamic acid	7.88 ± 2.21 ^ns^	9.02 ± 1.52	7.89 ± 1.08	8.21 ± 1.57	9.79 ± 1.53	11.9 ± 1.43	0.4958	0.5611	0.7041
Proline	3.19 ± 1.05 ^ns^	3.76 ± 0.48	3.58 ± 0.96	3.72 ± 0.96	3.81 ± 1.04	4.09 ± 0.35	0.9851	0.6667	0.8465
Glycine	1.97 ± 0.66 ^ns^	2.25 ± 0.43	1.98 ± 0.32	2.25 ± 0.56	2.56 ± 0.42	2.88 ± 0.32	0.7168	0.4339	0.7348
Alanine	8.84 ± 2.16 ^ns^	10.4 ± 0.9	8.93 ± 1.28	10.3 ± 1.4	11.7 ± 0.9	12.01 ± 2.4	0.4278	0.2031	0.6290
Valine	3.23 ± 0.76 ^ns^	3.46 ± 0.50	3.08 ± 0.51	4.36 ± 0.72	3.68 ± 0.41	4.10 ± 0.28	0.5674	0.3223	0.9414
Cystine	3.35 ± 1.40 ^ns^	3.73 ± 1.26	4.38 ± 1.41	3.20 ± 1.20	2.79 ± 0.82	5.30 ± 0.83	0.6894	0.6598	0.4488
Methionine	8.95 ± 3.45 ^ns^	10.5 ± 1.7	9.45 ± 1.52	9.10 ± 2.11	11.8 ± 2.11	14.0 ± 1.8	0.5753	0.5548	0.7290
Isoleucine	2.15 ± 0.38 ^ns^	3.00 ± 0.30	2.44 ± 0.87	3.22 ± 0.26	3.09 ± 0.16	4.09 ± 0.42	0.1227	0.1711	0.7186
Leucine	14.0 ± 5.3 ^ns^	14.4 ± 2.3	13.3 ± 1.4	14.8 ± 2.9	16.7 ± 1.9	20.1 ± 2.3	0.6285	0.5441	0.6209
Tyrosine	9.50 ± 3.92 ^ns^	10.9 ± 2.0	10.3 ± 1.7	10.5 ± 2.7	13.0 ± 2.6	14.0 ± 2.3	0.8135	0.4445	0.7664
Phenylalanine	6.14 ± 2.62 ^ns^	5.99 ± 1.35	5.52 ± 0.68	7.68 ± 2.58	7.15 ± 1.09	11.6 ± 1.3	0.2286	0.5192	0.7831
γ-aminobutyric acid	11.8 ± 2.4 ^ns^	13.5 ± 1.5	13.0 ± 2.1	14.8 ± 2.0	17.5 ± 1.7	16.9 ± 0.3	0.2481	0.0563	0.5308
Histidine	3.79 ± 1.33 ^ns^	4.41 ± 0.91	3.90 ± 0.65	4.41 ± 0.86	4.91 ± 0.78	5.76 ± 0.49	0.6436	0.4346	0.8170
Ornithine	10.6 ± 4.8 ^ns^	11.7 ± 2.4	11.0 ± 1.8	10.3 ± 2.9	16.6 ± 2.3	19.6 ± 2.4	0.2016	0.2725	0.3645
Lysine	36.2 ± 10.1 ^ns^	40.8 ± 4.6	34.6 ± 4.6	36.5 ± 4.4	48.8 ± 7.7	55.2 ± 4.1	0.2080	0.3157	0.3395
Ammonia	9.97 ± 2.37 ^ns^	8.86 ± 0.74	7.07 ± 0.56	7.92 ± 0.41	10.1 ± 1.5	12.3 ± 0.7	0.1219	0.8760	0.0694
Arginine	31.6 ± 7.9 ^ns^	34.4 ± 1.96	33.4 ± 5.0	36.6 ± 4.4	45.3 ± 4.9	46.7 ± 3.2	0.2056	0.0808	0.4015

^1^ Values are means (±SEM) from triplicate groups, N = 3. ^ns^: no significant difference.

**Table 5 animals-15-00809-t005:** Effects of temperature and GABA inclusion on plasma metabolites of juvenile olive flounder, followed by the acute temperature exposure test ^1^.

Temperature (°C)	Diet	GOT ^2^ (U/L)	GPT ^3^ (U/L)	GLU ^4^ (mg/dL)	TG ^5^ (mg/dL)	TP ^6^ (mg/dL)	TCHO ^7^ (g/L)
Interactive effects between diet and temperature							
19.5	GABA70	16.7 ± 1.2 ^ns^	13.3 ± 0.7 ^ns^	12.0 ± 1.7 ^ns^	241 ± 21.9 ^ns^	3.10 ± 0.10 ^ns^	161 ± 8 ^ns^
	GABA174	15.3 ± 0.3	14.0 ± 0.6	12.7 ± 0.3	194 ± 22.8	3.10 ± 0.10	161 ± 6
	GABA275	17.7 ± 2.4	16.7 ± 2.3	15.0 ± 2.1	273 ± 50	2.83 ± 0.10	159 ± 8
	GABA396	14.3 ± 0.9	13.0 ± 0.6	13.0 ± 0.6	230 ± 28	2.77 ± 0.10	137 ± 3
	GABA476	19.0 ± 2.5	14.3 ± 1.2	12.7 ± 0.3	237 ± 9.7	2.87 ± 0.10	140 ± 4
	GABA516	19.7 ± 2.6	15.3 ± 2.3	14.3 ± 0.3	271 ± 35	3.00 ± 0.10	171 ± 13
29	GABA70	43.7 ± 8.2	27.0 ± 2.0	50.2 ± 16.8	241 ± 26	3.10 ± 0.10	160 ± 7
	GABA174	50.3 ± 21.1	35.0 ± 8.6	116 ± 19	263 ± 13	3.23 ± 0.20	154 ± 8
	GABA275	37.3 ± 7.4	28.3 ± 2.3	64.3 ± 11.6	233 ± 28	3.07 ± 0.00	155 ± 8
	GABA396	28.7 ± 7.7	26.7 ± 6.7	33.0 ± 6.2	295 ± 7	3.30 ± 0.40	160 ± 12
	GABA476	33.0 ± 3.8	32.7 ± 8.2	77.0 ± 15.2	236 ± 21	3.33 ± 0.40	156 ± 2
	GABA516	85.0 ± 16.8	31.3 ± 3.3	72.2 ± 5.6	250 ± 21	3.23 ± 0.10	168 ± 11
Main effects of temperature							
19.5		17.1 ± 0.8 ^b^	14.4 ± 0.6 ^b^	13.3 ± 0.5 ^b^	241 ± 12 ^ns^	2.94 ± 0.00 ^b^	155 ± 4 ^ns^
29		46.3 ± 6.2 ^a^	30.2 ± 2.1 ^a^	68.7 ± 7.7 ^a^	253 ± 9	3.21 ± 0.1 ^a^	159 ± 3
Main effects of diet							
	GABA70	30.2 ± 7.1 ^ns^	20.2 ± 3.2 ^ns^	31.1 ± 11 ^ns^	241 ± 15 ^ns^	3.10 ± 0.1 ^ns^	161 ± 5 ^ns^
	GABA174	32.8 ± 12.3	24.5 ± 6.1	64.2 ± 24.6	228 ± 20	3.15 ± 0.1	158 ± 6
	GABA275	27.5 ± 5.6	22.5 ± 3.0	39.7 ± 12.2	253 ± 27	2.95 ± 0.1	157 ± 5
	GABA396	21.5 ± 4.7	19.8 ± 4.0	23.0 ± 5.3	263 ± 20	3.03 ± 0.2	148 ± 8
	GABA476	26.0 ± 3.7	23.5 ± 5.5	44.8 ± 15.9	237 ± 10	3.10 ± 0.2	148 ± 4
	GABA516	52.3 ± 16.5	23.3 ± 4.0	43.3 ± 13.2	260 ± 19	3.12 ± 0.1	170 ± 8
Two-way ANOVA (*p*-values)							
Temperature		<0.0001	<0.0001	<0.0001	0.4336	0.0235	0.4408
Diet		0.0517	0.8492	0.061	0.7289	0.923	0.1455
Temperature × Diet		0.6700	0.8932	0.060	0.2222	0.7647	0.4106

^1^ Values are means (±SEM) from triplicate groups (N = 3) of fish, where the values in each row with different superscripts are significantly different (*p* < 0.05); ^2^ glutamic oxaloacetic transaminase; ^3^ glutamic pyruvate transaminase; ^4^ glucose; ^5^ triglyceride; ^6^ total protein; ^7^ total cholesterol; ns: no significant difference; ^ns^: no significant difference.

Table 6 presents the results for the antioxidants, immune response, and stress hormone levels following the acute temperature exposure test. The interaction effects of GABA and temperature on glutathione peroxidase (GPx), superoxide dismutase (SOD), immunoglobulin M (IgM), lysozyme (LZM), and cortisol (CORT) were not significant across all experimental groups. Similarly, temperature had no significant main effect on GPx, IgM, and LZM concentrations; however, it exerted a significant main effect on SOD and CORT levels. Specifically, SOD levels at 29 °C were significantly lower (2.29 ± 0.36 µg/mL) compared to those at 19.5 °C (3.34 ± 0.17 µg/mL). Additionally, cortisol levels significantly increased at 29 °C (6395 ± 194 ng/mL) compared to those at 19.5 °C (5001 ± 147 ng/mL). The main effects of GABA on these measured parameters were not significant at any of the inclusion levels.

#### 3.2.4. Effects of Temperature and Dietary GABA on Molecular Response

The gene expression levels of the stress-related parameters in the livers and brains of juvenile olive flounder fed the experimental diets and subjected to acute temperature stress are detailed in Figure 4 and Figure 5. The mRNA expression levels of four key stress-related genes, including 60 kDa of heat shock protein (*hsp60*), 70 kDa of heat shock protein (*hsp70*), 90 kDa of heat shock protein (*hsp90*), and 65 kDa of warm temperature acclimation-related protein (*wap65*), were measured. The results indicated no significant differences in the expression levels of these genes among the different dietary groups in response to the stress.

## 4. Discussion

This study aimed to investigate whether GABA, as a dietary inclusion, could mitigate the effects of acute temperature stress in juvenile olive flounder, against the backdrop of its growing reputation for mitigating stress in both aquatic and non-aquatic species. We assessed its efficacy at various biological levels. At the whole-organism level, olive flounder juveniles fed the test diets showed no significant differences in any the growth performance parameters evaluated, namely, FBW, WG, SGR, FE, and FCR, as the levels were relatively unchanged in all treatment groups compared to those of the control (*p* > 0.05). This suggests that GABA did not enhance growth under normal temperature conditions. These results suggest that GABA inclusion did not significantly enhance growth performance compared to that of the control group. Lee et al. [7] found that GABA promoted growth in olive flounder juveniles reared under chronic temperature stress for 28 days. However, in the same study, GABA showed no significant growth effects under normal temperature conditions. This suggests that GABA’s growth-promoting effects may depend on the rearing conditions, as Abdel-Tawwab [47] reported that rearing conditions influence fish performance. Moreover, some studies have found that GABA inclusion did significantly improve growth performance in olive flounder reared under normal conditions [48]. However, under chronic density stress, GABA-treated olive flounder exhibited significantly enhanced growth parameters compared to those of the control group [49], suggesting that the efficacy of GABA may be more pronounced under chronic stress scenarios rather than under acute stress. In contrast, in white-leg shrimp reared under normal conditions, GABA inclusion significantly improved growth performance [50], whereas Bae et al. [49] reported no significant differences in growth parameters in olive flounder raised under normal culture conditions. Additionally, GABAergic systems in goldfish were reported to be temperature-sensitive [51], which could depend on the duration of exposure, probably attributed to a highly diverse repertoire of GABAA receptors [52].

Contrary findings have been reported in both olive flounder and other aquatic species, even under normal rearing conditions. Kim et al. [48] observed improved growth performance in olive flounder fed dietary GABA and reared at normal water temperatures, although their experimental fish displayed a much smaller initial mean weight of 0.4 g, which may explain the discrepancy. Similarly, Farris et al. [29] reported significant growth enhancement in olive flounder with an initial weight of 4.90 g, which was also considerably smaller than those used in our study. These contrasting results suggest that GABA’s effects on growth performance may be size-dependent. Further research is needed to clarify the extent to which fish size influences GABA-mediated growth responses. Other species in which GABA significantly improved growth performance, even under normal rearing conditions, include Nile tilapia (*Oreochromis niloticus*), whiteleg shrimp, and grass carp (*Ctenopharyngodon idella*) [29,30,53,54,55], additionally suggesting species-specific responses as a critical factor influencing the growth-inducing effects of GABA. These observations underscore the importance of considering the size of fish, species specificity, and environmental conditions (normal or stressed) in GABA-related aquaculture strategies.

In the analysis of the whole-body composition at the end of the trial, it was found that dietary GABA inclusion did not affect the whole-body composition of olive flounder. Similar observations regarding the effects of GABA inclusion on whole-body composition have been reported in whiteleg shrimp, juvenile olive flounder, juvenile grass carp, and Nile tilapia [29,30,49,51,53,56,57]. Additionally, the free amino acid levels in the brain, intestine, and plasma were also unaffected after the feeding trial. Nutrients dissolved in plasma are transported to tissues via capillaries, where they can cross the capillary walls through mechanisms such as diffusion, filtration, and osmosis [54]. In this study, the assessment of the plasma-free amino acid (FAA) profiles in olive flounder did not reveal significant differences, suggesting that plasma amino acid homeostasis remained balanced and was not significantly affected by GABA inclusion. This stability in plasma-FAA profiles may be due to an adequate dietary protein supply, ensuring sufficient availability of essential amino acids for physiological processes, as dietary protein deficiency has been reported to lead to the utilization of essential amino acids and non-essential amino acids to meet metabolic demands [49], thereby altering the plasma-free amino acid profile [58]. Additionally, plasma amino acid profiles in rainbow trout were reported to be influenced by different dietary nutrient conditions [59].

The adaptability of the GABAergic system may effectively mitigate the activation of the HPI axis in fish, enabling the brain to dynamically respond to environmental stressors [60,61,62]. This adaptability helps maintain neural stability and supports the organism’s ability to cope with stress by inducing a cascade of physiological and metabolic adjustments aimed at preserving homeostasis [61,62,63]. Collectively, these mechanisms contribute to sustaining survival in fish exposed to stress [64]. In our assessment of GABA inclusion under lethal temperature stress, the hazard risk in the GABA516 group plateaued the earliest at the 30 h mark postexposure, with no further mortality observed thereafter. This suggests that higher GABA inclusion may have temporarily enhanced the adaptability of the GABAergic system, providing some protection against extreme temperature stress. In contrast, mortality in the GABA174 group persisted beyond the 30 h mark and continued until the experiment’s conclusion, indicating the lowest resistance to stress among all groups. This could be due to insufficient activation of the GABAergic system’s adaptive mechanisms under extreme temperature stress. However, despite these trends, Kaplan–Meier survival analysis revealed no significant differences in survival rates among all groups by the end of the exposure period. This finding suggests that while high dietary GABA inclusion initially conferred a protective advantage, prolonged exposure to extreme temperatures likely overwhelmed the physiological benefits, suggesting that the protective effects of GABA on olive flounder under extreme-temperature conditions may be time-bound.

Blood parameters can reflect the health condition and nutritional metabolism of fish, making them useful for determining the health status of fish in response to dietary additions [53]. Different blood parameters usually change according to stress conditions and various other environmental factors [65,66,67]. We assessed the interaction and main effects of GABA inclusion and acute temperature stress on the plasma metabolites. Glutamate oxaloacetate transaminase (GOT) and glutamate pyruvate transaminase (GPT) are liver enzymes that serve as key indicators of liver health and metabolic activity, particularly in response to environmental stressors. Elevated levels of these enzymes typically signal hepatic damage or increased metabolic demand, both of which are associated with temperature stress [67,68,69]. In the present study, there was no interaction effect between temperature and diet on these enzymes. While no interaction effect between temperature and diet was observed, temperature alone significantly increased GOT and GPT levels (*p* < 0.05). Similar increases in these enzymes have been reported in aquaculture studies under various stressors [70]. The observed elevations in GOT and GPT levels may indicate stress-induced tissue damage or impaired liver function [71,72]. Interestingly, while increased GOT and GPT levels are often considered hallmarks of hepatic stress [73], some studies suggest that these elevations may be species-specific or dependent on the severity of the stressor. For instance, Dawood et al. [74] found that ammonia stress led to increased GOT and GPT levels in common carp, a trend also observed under transportation-induced stress [75]. However, in rohu (*Labeo rohita*) exposed to chemical stress, both enzymes remained unaffected up until after 48 h of exposure to chemical toxicity stress [76]. This disparity suggests the species-specific stress adaptation mechanisms may influence enzymatic responses to stress [77,78]. Moreover, the physiological response to stress varies, depending on multiple factors, including species, sex, and age, as well as the type and duration of the stressor [74].

Glucose (GLU) is a critical marker of energy metabolism and stress response, as GLU levels often rise during stress to meet the organism’s increased energy requirements [79,80]. Our study found no significant interaction effects between temperature and diet on GLU levels, aligning with the findings of Bae et al. [49], who reported no interaction effects in olive flounder exposed to density stress following GABA inclusion. However, we observed a significant main effect of temperature, indicating its detrimental impact on glucose metabolism in this species. The increase in GLU levels observed in this study is likely driven by glycogen breakdown and gluconeogenesis, both of which are essential metabolic responses to stress. These processes are primarily regulated by catecholamines and cortisol, which play crucial roles in mobilizing energy reserves to help fish cope with environmental challenges [81,82].

Our results are consistent with those of Zhang et al. [83], who found that GABA did not alter hemolymph glucose concentrations in juvenile Chinese mitten crabs (*Eriocheir sinensis*) after six hours of fasting stress. However, prolonged stress exposure significantly reduced glucose levels compared to those of the control, suggesting that GABA’s role in glucose metabolism may depend on the duration of stress exposure. Similarly, Bae et al. [49] reported that density stress significantly affected GLU levels in olive flounder, implying that GLU’s role as an energy marker may vary across species and stressor types. In contrast, GABA inclusion was effective in mitigating stress in bottom-dwelling *Cirrhinus mrigala* subjected to hypoxia stress [84]. Similarly, Temu et al. [56] reported an increased GLU concentration in Nile tilapia. This discrepancy may stem from species-specific metabolic differences or the nature of the stressor, highlighting the need for a deeper understanding of GLU as a stress biomarker in fish.

Furthermore, we observed that GABA inclusion did not significantly affect plasma cortisol levels; however, exposure to elevated temperatures resulted in a significant increase in cortisol concentrations. This finding is consistent with previous reports on olive flounder. For instance, Bae et al. [49] found that GABA inclusion had no effect on cortisol levels in flounder raised at low, normal, and high stocking densities. However, density itself had a significant effect, with higher densities leading to increased cortisol levels, indicating that stress influenced the regulation of this hormone. Similarly, in our experiment, while GABA inclusion did not alter cortisol levels, temperature had a significant impact, causing an increase in cortisol concentrations.

Total protein (TP) in the blood is a critical indicator of overall health and nutritional status in fish, as it reflects the balance between protein synthesis and degradation [85]. In this study, no significant interaction effects on TP levels were observed. However, similar to the trends noted in the GOT, GPT, and GLU levels, temperature emerged as a critical factor, significantly increasing TP levels from 2.94 ± 0.00 mg/dL to 3.21 ± 0.1 mg/dL. This suggests that elevated temperatures influence protein metabolism and physiological responses to stress. The elevated TP levels suggest hyperproteinemia, a metabolic condition that can have complex deleterious effects on homeostasis [86]. Prolonged exposure to stressors can therefore induce physiological strain, weaken immune defenses, and reduce growth efficiency [87], thereby posing a threat to fish well-being [88]. Moreover, this increase in TP levels may indicate a heightened demand for protein synthesis due to the global inhibition of protein synthesis in response to stress [89] or protein mobilization as a compensatory mechanism to mitigate stress-induced damage. In Nile tilapia, serum TP levels were not significantly affected by density stress when the fish were reared in a biofloc environment [90], highlighting the potential role of environmental conditions to modulate TP responses to stress. Similarly, TP remained unaffected by GABA when juvenile red seabream (*Pagrus major*) were subjected to nutritional stress [91]. Understanding these biological responses is essential for developing strategies to mitigate their impact on aquaculture [92,93].

Furthermore, we assessed triglycerides (TG) and total cholesterol (TCHO) as markers of lipid metabolism [93,94]. The levels of these markers were not influenced by diet, temperature, or their combination, suggesting that lipid metabolism remained stable under the experimental conditions. This indicates a balanced physiological state in terms of lipid synthesis, mobilization, and utilization. Similar findings were reported by Varghese et al. [83], in which GABA did not significantly alter TG and TCHO levels in bottom-dwelling carp subjected to hypoxia stress, as well as in juvenile red seabream [91]. This highlights the complexity of lipid metabolism responses and raises the question of whether GABA inclusion can exert a more pronounced effect on lipid metabolism under specific environmental stressors.

SOD and GPx are key antioxidant enzymes that mitigate oxidative stress by neutralizing reactive oxygen species generated during thermal stress, thereby serving as indicators of oxidative damage and cellular defense mechanisms [95]. IgM and LZM were included as critical components of the innate immune system, which is responsible for the primary immune defense against pathogens, particularly in stressed fish [90]. CORT is a well-known biomarker of the stress response in fish; its levels typically rise during acute environmental stress, providing insight into the physiological stress response [96]. By evaluating these parameters, we aimed to comprehensively assess the influence of GABA inclusion on the antioxidant defense and immune response systems under acute temperature stress conditions. We analyzed the effects of a GABA-supplemented diet and temperature on the plasma levels of SOD, GPx, IgM, LZM, and CORT. Our results showed no significant interaction or main effects of the diet on measures of the immune system, antioxidants, or stress hormones. However, temperature exerted significant effects on both SOD and CORT levels, indicating that temperature stress results in disturbances in these metabolites [97].

Although dietary treatments did not significantly influence immune, antioxidant, or stress hormone parameters, our findings emphasize the critical role of temperature in modulating SOD and CORT levels. Increased SOD activity suggests an upregulation of antioxidant defenses to mitigate the oxidative damage caused by temperature-induced stress [98]. This response is essential for maintaining cellular integrity but may indicate an elevated oxidative burden that could impair fish health, if prolonged [99]. Similarly, elevated CORT levels, a key indicator of hypothalamic–pituitary–interrenal (HPI) axis activation, suggest heightened physiological stress in fish exposed to environmental challenges [100]. This stress response is a critical part of the survival mechanism of the fish, enabling them to adapt to adverse conditions [101]. However, a long-term elevation in CORT can lead to chronic stress and its resulting consequences, negatively impacting the health of the fish [102]. Prolonged high levels of CORT can suppress immune function [103], leaving fish more vulnerable to infections and disease outbreaks [104]. This immunosuppression can increase mortality rates and lead to considerable economic losses [105].

Our findings align with those of previous studies in other species, including common carp (*Cyprinus carpio*) [106], olive flounder [7], Indian major carp (*Labeo rohita*) [107,108], and Nile tilapia (*Oreochromis niloticus*) [109], in which temperature-induced stress led to similar physiological responses. This sensitivity highlights the role of temperature increase in triggering physiological responses, including the stress axis [110].

Furthermore, Xie et al. [31], in consonance with our findings, also reported no significant changes in catalase activity following ammonia stress, but they found that total SOD levels in the hemolymph and hepatopancreas were affected by stress. The absence of temperature effects on the other enzymes in our study suggests that distinct regulatory mechanisms govern their responses to environmental stimuli [111]. These differential responses may arise from the diverse roles these enzymes play as the first line of defense in the system stress adaptation pathways [112]. Further research is necessary to elucidate the mechanisms driving these observed patterns and to identify the specific environmental factors influencing enzyme activity in olive flounder.

At the molecular level, the expression of stress-related genes, including *hsp60*, *hsp70*, *hsp90*, and *wap65*, in the brain and liver showed no significant changes across dietary groups. This indicates that GABA inclusion did not induce a differential stress response in the tissues analyzed. Our findings are consistent with those of Lee et al. [7], who reported that *hsp70* and *hsp90* levels remained unchanged in olive flounder fed GABA-supplemented diets under high water-temperature conditions. However, contrasting results have been observed in previous studies, such as that of Lin et al. [113], in which a significant upregulation of *hsp70* and *hsp90* was reported in largemouth bass (*Micropterus salmoides*) exposed to the combined stressors of flow and density, and in Chinese mitten crab (*Eriocheir sinensis*) exposed to hypoxic stress [114]. According to Kregel [115], there are various physiological factors that modulate *hsp* responses to stressors at the cellular and systemic levels. Additionally, factors such as species specificity play a role in GABAergic signaling, as species- and anatomic-specific differences exist in receptor pharmacology [116]. Further research is needed to clarify the conditions under which GABA inclusion may influence the expression of stress-related genes in different aquaculture species.

Furthermore, in response to stress stimuli, an adaptive and protective response is triggered, which depends on the duration of and exposure to the stress [117]. Studies in species such as zebrafish and Atlantic salmon revealed similar trends [118,119], emphasizing the conserved nature of these stress-response pathways across teleost fish. This underscores the need to explore species-specific stress tolerance thresholds [120], which could guide the development of management strategies to mitigate the effects of climate variability on aquaculture productivity. The mechanisms underlying the lack of differential expression observed in this study require further investigation to elucidate the complex interactions between GABA inclusion and stress response pathways.

The effects of GABA on stress regulation are highly context-dependent, influenced by factors such as the type, intensity, and duration of the stressor, as well as the fish’s physiological state. This variability underscores the complex interaction between GABAergic signaling and stress-response pathways, indicating that GABA’s role in modulating stress is not consistent or predictable across different conditions, while GABA functions as an inhibitory neurotransmitter, reducing hyperactivation of the hypothalamic–pituitary–interrenal (HPI) axis, a key regulator of stress in teleost fish [48]. Its effectiveness appears to be influenced by environmental factors such as temperature, salinity, and oxygen availability, all of which affect the HPI axis and related pathways. Additionally, a limitation of this study was the short duration of GABA inclusion, which may not have been long enough to produce measurable effects on stress-related parameters. Future research should explore longer inclusion periods and varying stress durations to fully assess GABA’s potential for stress mitigation. Furthermore, investigating the species-specific responses to GABA, along with factors like age, size, and genetic background, would provide a more comprehensive understanding of GABA’s applicability across different aquaculture settings.

Understanding these mechanisms is essential for developing strategies to enhance aquaculture resilience, particularly in the face of climate change and environmental fluctuations. Identifying the conditions under which GABA inclusion is most effective could form the basis for dietary interventions designed for specific stress scenarios.

Taken together, our findings indicate that dietary GABA inclusion did not provide observable protective effects for olive flounder under acute temperature stress. However, our results underscore the significant health challenges faced by olive flounder due to rising temperature stress, which continues to impact flounder farms. The detrimental effects of temperature stress on this species are a growing concern, particularly as climate change intensifies.

According to a United Nations report [121], the ocean absorbs 90% of the heat generated by global warming, raising serious concerns about the sustainability of fish species like olive flounder, whose aquaculture operations heavily rely on direct ocean water. In response to these concerns, researchers are increasingly focused on understanding olive flounder mortality associated with rapid temperature fluctuations in aquaculture environments [122]. In a survey on welfare-related issues affecting olive flounder farms in Korea, Oh and Kee [123] reported that water temperature fluctuations were identified as a critical challenge by 81.7% of respondents.

Given these pressing needs, this study evaluated the effects of dietary gamma-aminobutyric acid (GABA) inclusion on acute temperature stress in olive flounder. While GABA did not show observable effects in mitigating temperature stress, the findings highlight key physiological and metabolic responses to temperature fluctuations. This underscores the need for further research into alternative nutritional strategies to enhance stress resilience in aquaculture species.

## 5. Conclusions

This study serves as a preliminary investigation into the potential of dietary GABA inclusion to influence the stress response of juvenile olive flounder under acute temperature stress, based on GABA’s well-documented role as a stress mitigant across various species and stress conditions. Despite previous evidence supporting its benefits, the inclusion of GABA in the diet did not yield significant improvements in growth performance, body composition, or free amino acid profiles in olive flounder. Additionally, no significant changes were observed in survival rates following exposure to lethal temperature stress, suggesting that GABA inclusion did not enhance resilience to acute thermal stress in this species. However, elevated temperatures negatively impacted key physiological and metabolic parameters, including GOT, GPT, GLU, TP, SOD, and CORT, highlighting the susceptibility of olive flounder to temperature fluctuations, particularly during the summer months. These findings emphasize the need for effective dietary protocols to mitigate temperature-induced stress and the importance of further research on GABA’s potential applications in aquaculture. While this study did not provide conclusive evidence for GABA’s role in improving stress resilience under acute thermal stress, its potential as a mitigatory agent should not be dismissed. Given its established reputation in regards to stress regulation, additional research is necessary to optimize inclusion protocols and determine the specific conditions under which GABA may offer greater benefits. These insights could contribute to the development of more effective strategies for improving fish health and welfare in aquaculture systems, particularly in the face of a rapidly changing climate.

## Figures and Tables

**Figure 1 animals-15-00809-f001:**
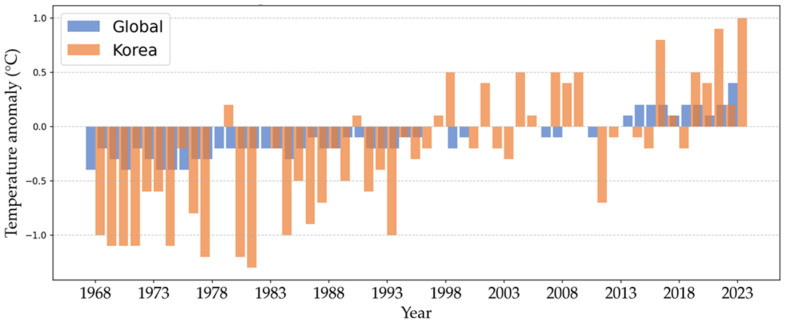
Rate of change in sea surface temperature anomalies in Korea compared to the global average (1968 to 2023). Source: National Institute of Fisheries Science in the Republic of Korea [5].

**Figure 2 animals-15-00809-f002:**
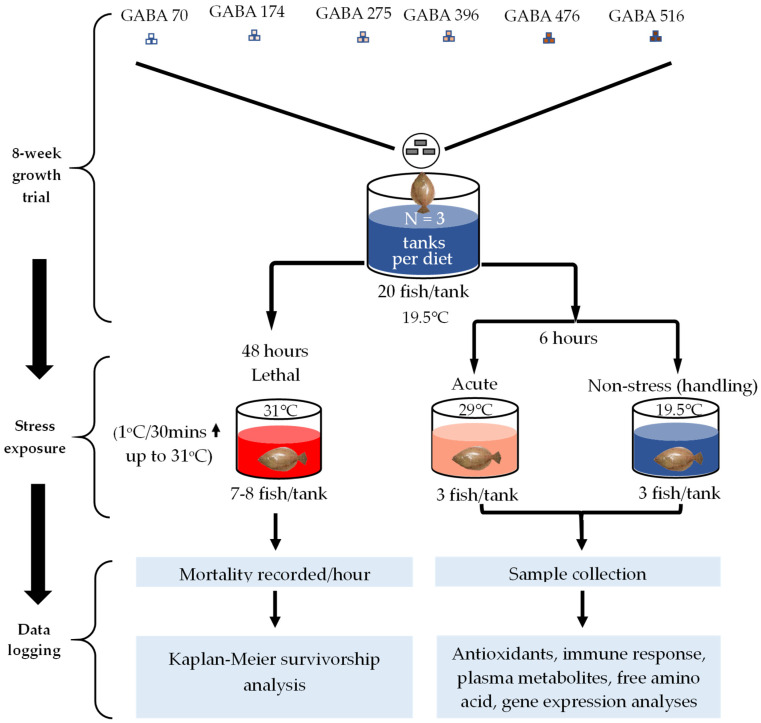
Experimental design of the feeding trial and temperature stress exposure tests.

**Figure 3 animals-15-00809-f003:**
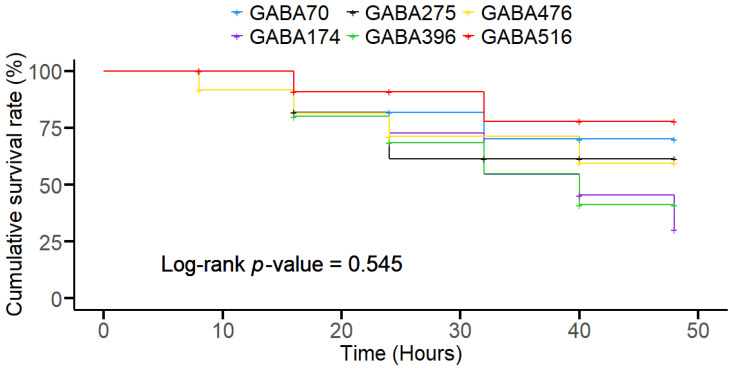
Kaplan–Meier survivorship curves (cumulative survival %; N = 3) showing the percentage of olive flounder survival over 48 h under acute temperature stress following GABA inclusion at various doses. The different GABA treatment groups include GABA70 (red), GABA174 (yellow), GABA275 (green), GABA396 (blue), GABA476 (light blue), and GABA516 (pink). Survival rates were monitored every hour. The survival rate average was not significantly different, with an average of 28.5 ± 4.6% (*p* > 0.05) at the end of the exposure period.

**Figure 4 animals-15-00809-f004:**
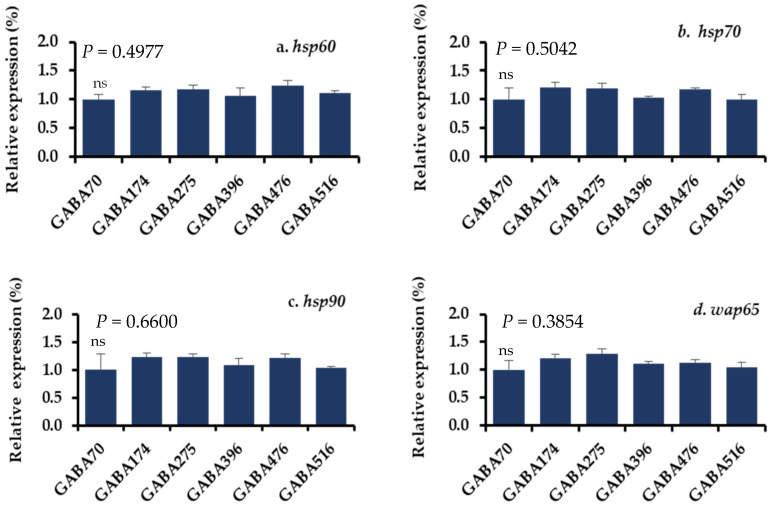
(**a**–**d**) Relative gene expression levels in the livers of olive flounder after the acute temperature exposure test. A vertical line with an error bar indicates the mean ± SEM for each treatment (N = 3). *hsp60*: heat shock protein 60 kDa; *hsp70*: heat shock protein 70 kDa; *hsp90*: heat shock protein 90 kDa; *wap65*: Warm temperature acclimation-related protein 65 kDa; ns: no significant difference.

**Figure 5 animals-15-00809-f005:**
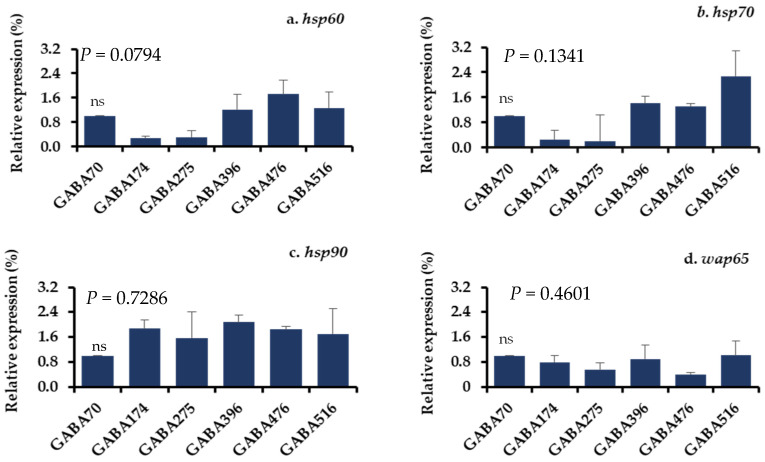
(**a**–**d**) Relative gene expression levels in the brain of olive flounder after the acute temperature exposure test. A vertical line with an error bar indicates the mean ± SEM for each treatment (N = 3). *hsp60*: heat shock protein 60-kDa; *hsp70*: heat shock protein 70-kDa; *hsp90*: heat shock protein 90 kDa; *wap65*: warm temperature acclimation-related protein 65 kDa; ns: no significant difference.

**Table 1 animals-15-00809-t001:** Feed formulation and proximate composition of the experimental diets.

Ingredients (%)	Diets					
	GABA70	GABA174	GABA275	GABA396	GABA476	GABA516
Anchovy fishmeal ^1^	20	20	20	20	20	20
Starch ^1^	5	5	5	5	5	5
Wheat flour ^1^	13.7	13.7	13.7	13.7	13.7	13.7
Squid liver powder ^1^	5	5	5	5	5	5
Soybean meal ^1^	9	9	9	9	9	9
Poultry by-product ^1^	9.5	9.5	9.5	9.5	9.5	9.5
Isolated soybean protein ^1^	9.5	9.5	9.5	9.5	9.5	9.5
Tankage meal ^1^	14	14	14	14	14	14
Fish oil ^1^	5.5	5.5	5.5	5.5	5.5	5.5
Lecithin ^1^	0.9	0.9	0.9	0.9	0.9	0.9
Betaine ^1^	0.9	0.9	0.9	0.9	0.9	0.9
Taurine ^1^	0.9	0.9	0.9	0.9	0.9	0.9
Monocalcium phosphate ^2^	0.9	0.9	0.9	0.9	0.9	0.9
Methionine ^1^	0.4	0.4	0.4	0.4	0.4	0.4
Lysine ^1^	0.4	0.4	0.4	0.4	0.4	0.4
Mineral mix ^3^	1.2	1.2	1.2	1.2	1.2	1.2
Vitamin mix ^4^	1.2	1.2	1.2	1.2	1.2	1.2
Vitamin C ^1^	0.2	0.2	0.2	0.2	0.2	0.2
Choline	0.8	0.8	0.8	0.8	0.8	0.8
Cellulose ^1^	1	0.8	0.6	0.4	0.2	0
GABA premix	0	0.2	0.4	0.6	0.8	1
Total	100	100	100	100	100	100
Proximate composition ^5^ (%)						
Dry matter	98.4 ± 0.0	98.7 ± 0.0	99.2 ± 0.0	99.0 ± 0.0	98.2 ± 0.0	98.4 ± 0.0
Crude protein	53.47 ± 0.28	53.2 ± 0.09	54.38 ± 0.20	53.5 ± 0.20	53.05 ± 0.15	53.71 ± 0.19
Crude lipid	12.13 ± 0.03	12.07 ± 0.12	11.88 ± 0.02	11.57 ± 0.19	11.89 ± 0.15	11.92 ± 0.00
Crude ash	10.12 ± 0.07	9.87 ± 0.04	9.99 ± 0.09	9.83 ± 0.04	9.88 ± 0.06	9.96 ± 0.08
Gross energy (kcal/kg)	5149 ± 6	5174 ± 49	5097 ± 36	5130 ± 12	5150 ± 28	5135 ± 0

^1^ The Goyang feed company, Goyang-si, Republic of Korea. ^2^ Duksan Pure Chemicals Co., Ltd., Ansan-si, Republic of Korea. ^3^ Mineral mix (g/kg premix): ferrous fumarate, 12.50; manganese sulfate, 11.25; dried ferrous sulfate, 20.0; dried cupric sulfate, 1.25; dobaltous sulfate, 0.75; zinc sulfate KVP, 3.75; calcium iodate, 0.75; magnesium sulfate, 80.20; aluminum hydroxide, 0.75. ^4^ Vitamin mix (mg/kg premix): A, 1,000,000 IU; D, 200,000 IU; E, 10,000; B1, 2000; B6, 1500; B12, 10; C, 10,000; calcium pantothenic acid, 5000; nicotinic acid 4500; B-biotin 10; Choline chloride, 30,000; inositol, 5000. ^5^ The proximate composition analysis was performed in duplicate for each of the diets.

**Table 2 animals-15-00809-t002:** List of primers used for gene expression.

Genes	Primer Sequences	Tm (°C)	Accession Number	Product Size (bp)	Efficiency (%)	R^2^	Slope	Reference
*β-actin*	F: GGAATCCACGAGACCACCTACA	62.1	XM_020109620.1	264	99.9	0.9515	−3.3250	[41]
	R: CTGCTTGCTGATCCACATCTGC	62.1						
*hsp60* ^1^	F: TGACTTCGGGAAAGTCGGTG	59.3	XM_020105844.1	2927	100.1	0.8904	−3.3200	[42]
	R: ACGATCTCCAGTGCACGTTT	57.3						
*hsp70* ^2^	F: TTCAATGATTCTCAGAGGCAAGC	58.9	XM_020089177.1	113	99.4	0.9994	−3.3364	[43]
	R: TTATCTAAGCCGTAGGCAATCGC	60.6						
*hsp90* ^3^	F: GAGCGAGACAAGGAGGTGAG	61.4	XM_020091873.1	100	96.8	0.9944	−3.4016	[7]
	R: CTGGCTTGTCTTCGTCCTTC	59.3						
*wap65* ^4^	F: AACCAAGGCTGTGGAGAAGAAAGAG	63	XM_020105098.1	1727	98.6	0.9398	−3.3550	[44]
	R: GTGTCCGTGGAAGCAGTAGTAGTG	64.4						

^1^ Heat shock protein 60 kDa. ^2^ Heat shock protein 70 kDa. ^3^ Heat shock protein 90 kDa. ^4^ Warm temperature acclimation-related protein 65 kDa.

**Table 3 animals-15-00809-t003:** Growth response and body composition of juvenile olive flounder fed the experimental diets for 8 weeks ^1^.

Parameters ^2^	Diets						Pr > F		
	Control	GABA174	GABA275	GABA396	GABA476	GABA516	ANOVA	Linear	Quadratic
IBW (g)	13.4 ± 0.2 ^ns^	12.9 ± 0.1	12.9 ± 0.2	13.0 ± 0.3	13.0 ± 0.3	12.7 ± 0.1	0.4042	0.4253	0.1428
FBW (g)	48.6 ± 0.2 ^ns^	48.2 ± 0.3	49.7 ± 0.9	48.3 ± 0.8	49.2 ± 1.1	50.3 ± 0.6	0.3184	0.5901	0.9675
WG (%)	264 ± 7 ^ns^	275 ± 1	286 ± 1	273 ± 14	278 ± 12	296 ± 2	0.1629	0.3218	0.2341
SGR (%)	2.22 ± 0.03 ^ns^	2.28 ± 0.00	2.33 ± 0.00	2.27 ± 0.07	2.29 ± 0.06	2.37 ± 0.01	0.1736	0.298	0.2353
FE (%)	111 ± 17 ^ns^	129 ± 9	97.1 ± 26.4	99.2 ± 18.9	111 ± 18	109 ± 16	0.8499	0.621	0.7643
FCR	2.35 ± 0.07 ^ns^	2.11 ± 0.06	2.17 ± 0.04	2.06 ± 0.07	2.32 ± 0.12	2.16 ± 0.02	0.0664	0.6061	0.0730
SR (%)	100 ± 0 ^ns^	100 ± 3	100 ± 0	98.3 ± 1.7	102 ± 1.7	98.4 ± 1.6	0.7291	0.7554	0.4348
Body composition (%; as is)									
Moisture	75.0 ± 0.1 ^ns^	75.0 ± 0.2	75.0 ± 0.0	74.8 ± 0.2	75.0 ± 0.2	75.0 ± 0.2	0.9817	0.7512	0.8403
Crude protein	18.0 ± 0.0 ^ns^	18.1 ± 0.1	18.0 ± 0.1	18.2 ± 0.4	18.0 ± 0.3	17.8 ± 0.3	0.8367	0.8155	0.6211
Crude lipid	3.07 ± 0.14 ^ns^	3.11 ± 0.02	3.04 ± 0.08	2.89 ± 0.48	3.10 ± 0.09	3.15 ± 0.18	0.9689	0.8313	0.7738
Crude ash	3.80 ± 0.20 ^ns^	3.62 ± 0.10	3.62 ± 0.20	3.77 ± 0.14	3.75 ± 0.06	3.77 ± 0.04	0.8577	0.8949	0.3478

^1^ Values are means (±SEM) from triplicate groups (N = 3) of juvenile olive flounder; ^2^ IBW, initial body weight; FBW, final body weight; WG, weight gain; SGR, specific growth rate; FE, feed efficiency; FCR, feed conversion ratio; SR, survival rate. ^ns^: no significant difference.

**Table 6 animals-15-00809-t006:** Effects of temperature and GABA on plasma antioxidants enzymes, immune response, and stress hormones of juvenile olive flounder, followed by the acute temperature exposure test ^1^.

Temperature (°C)	Diet	GPx ^2^ (mU/mL)	SOD ^3^ (µg/mL)	IgM ^4^ (µg/mL)	LZM ^5^ (µg/mL)	CORT ^6^ (ng/mL)
Interactive effects between diet and temperature						
19.5	GABA70	21.3 ± 0.8 ^ns^	3.75 ± 0.29 ^ns^	4.43 ± 0.43 ^ns^	1.20 ± 0.22 ^ns^	4.95 ± 0.30 ^ns^
	GABA174	26.8 ± 5.1	3.69 ± 0.34	4.63 ± 0.42	1.08 ± 0.09	5.14 ± 0.30
	GABA275	23.5 ± 2.7	3.06 ± 0.25	4.61 ± 0.48	1.39 ± 0.37	4.84 ± 0.36
	GABA396	22.3 ± 1.7	3.22 ± 0.37	4.84 ± 0.05	1.01 ± 0.18	4.40 ± 0.44
	GABA476	22.4 ± 1.9	3.60 ± 0.29	5.34 ± 0.16	1.11 ± 0.14	5.32 ± 0.30
	GABA516	21.3 ± 3.7	2.71 ± 0.81	4.82 ± 0.33	0.85 ± 0.05	5.347 ± 0.445
29	GABA70	24.3 ± 6.4	3.55 ± 1.28	4.47 ± 0.25	1.30 ± 0.40	6.46 ± 0.51
	GABA174	18.9 ± 7.4	1.78 ± 0.67	4.79 ± 0.18	1.39 ± 0.36	6.61 ± 0.20
	GABA275	16.5 ± 3.8	1.26 ± 0.57	4.38 ± 0.49	1.34 ± 0.58	5.629 ± 0.386
	GABA396	18.7 ± 1.5	2.22 ± 0.28	4.91 ± 0.47	1.31 ± 0.15	6.29 ± 0.25
	GABA476	22.0 ± 9.4	2.36 ± 1.24	4.39 ± 0.15	1.00 ± 0.01	5.94 ± 0.36
	GABA516	25.8 ± 4.9	2.60 ± 1.04	4.58 ± 0.28	0.98 ± 0.10	7.29 ± 0.67
Main effects of temperature						
19.5		22.9 ± 1.1 ^ns^	3.34 ± 0.17 ^a^	4.78 ± 0.14 ^ns^	1.11 ± 0.08 ^ns^	5.00 ± 0.15 ^b^
29		21.0 ± 2.1	2.29 ± 0.36 ^b^	4.53 ± 0.12	1.22 ± 0.12	6.40 ± 0.19 ^a^
Main effects of diet						
	GABA70	22.8 ± 3.0 ^ns^	3.65 ± 0.59 ^ns^	4.45 ± 0.22 ^ns^	1.25 ± 0.21 ^ns^	5.71 ± 0.43 ^ns^
	GABA174	22.8 ± 4.4	2.73 ± 0.95	4.71 ± 0.08	1.23 ± 0.15	5.87 ± 0.73
	GABA275	20.0 ± 4.1	2.16 ± 0.49	4.50 ± 0.31	1.36 ± 0.31	5.234 ± 0.30
	GABA396	20.5 ± 3.5	2.72 ± 0.31	4.69 ± 0.22	1.16 ± 0.12	5.35 ± 0.48
	GABA476	22.2 ± 2.8	2.98 ± 0.63	4.86 ± 0.47	1.07 ± 0.07	5.63 ± 0.24
	GABA516	23.6 ± 2.9	2.66 ± 0.06	4.70 ± 0.12	0.92 ± 0.06	6.32 ± 0.97
Two-way ANOVA (*p*-values)						
Temperature		0.4819	0.0194	0.2306	0.4766	<0.0001
Diet		0.9604	0.4822	0.8734	0.6596	0.1291
Temperature × Diet		0.6727	0.7292	0.7484	0.9653	0.4991

^1^ Values are means (±SEM) from triplicate groups (N = 3) of fish, where the values in each row with different superscripts are significantly different (*p* < 0.05); ^2^ glutathione peroxidase; ^3^ superoxide dismutase; ^4^ immunoglobulin M; ^5^ lysozyme; ^6^ Cortisol; ^ns^: no significant difference.

## Data Availability

The data that support the findings of this study are available on request from the corresponding authors. The data are not publicly available due to privacy or ethical restrictions.

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
