# Peer review of "Effects of Dietary Gamma-Aminobutyric Acid (GABA) Inclusion on Acute Temperature Stress Responses in Juvenile Olive Flounder (Paralichthys olivaceus)"

_animals, 2025, doi:10.3390/ani15060809_

Round 1
Reviewer 1 Report (Previous Reviewer 4)
Comments and Suggestions for Authors
The manuscript has improved a lot when compared to other versions submitted previously; but there are still some details before publication.
1) In the abstract section, this type of sentence is unnecessary: ​​"Our findings highlight the need for further research to optimize GABA inclusion strategies, particularly with consideration for long-term physiological impacts." This type of information should be in the discussion section. Remove it.
2) I don't understand why you used the word sustainability as a keyword.
3) Our observations from this study will provide valuable insights into the physiological responses of olive flounder during temperature fluctuations and offer a clearer understanding of considerations necessary to utilize GABA as a dietary mitigant to enhance stress resilience in aquaculture species. Okay, that's your hypothesis; but what is the objective, clearly???
4) In the tables, there needs to be a scientific standard; with the numbers, for example in table 1; there is a variety of presentation. I recommend that you present only 3 numbers (or four if you think it is better and when necessary), as an example: xxx, xx.x or x.xx (in numbers it would be, for example, 258 or 25.8 or 2.58. This applies to all tables.
5) The discussion has improved a lot; but the conclusion is very difficult to evaluate without having a clear objective in the introduction section; I really don't understand that the conclusion should be separated into paragraphs; but rather a single paragraph, which can connect an idea based on the main and complementary results; in this form, it seems like it is more than one experiment. Check it out.
Author Response
Please see the attachment.

Reviewer 2 Report (Previous Reviewer 5)
Comments and Suggestions for Authors
This is a revised version of a previously rejected manuscript. This article deals with the effects of dietary GABA on growth of olive flounder, its survival after a sub-acute heat stress (48 h) and physiological responses to an acute heat stress (6 h).
There is a big question regarding the rationale of exposing fish to a short term heat stress, as chronic stress is an important issue in aquaculture. Also, it is strange the authors assessed the physiological responses after 6-h heat stress, not 48-h.
This is more surprising when I found that the authors were aware of that GABA is not a growth-promoter agent in flounder under normal rearing conditions.
There is a clear flaw in the methodology. When the effects of two factors (GABA and stress) are assessed on fish, all parameters must be measured both before and after stress. However, the authors did not do this. Important factors like amino acid profiles and trancription responses to heat stress were only assessed after the heat stress.
Proximate composition of the diets seems wrong. The authors stated it was dry matter basis. So, they should not report moisture! Moreover, how a diet have 1-2% moisture as dry matter basis?!!
The exact value of some parameters seems wrong. For example, blood glucose levels of fish is typically above 30 mg/dl. Blood cortisol in unstressed fish should be below 50 ng/ml and ideally below 20 ng/ml.
Discussion is lengthy and problematic. The first and second paragraphs should be merged and summarized.
L573-575: The authors should interpret the results based on what they found. Here they tried to express the fish in 517 mg/kg GABA had the lowest risk, however, the control group (70 mg/kg) also had a plateaued mortality after 30 h.
Line 598-603: The authors must avoid repeating the results in the discussion section. In these lines, they presented the results even with numerical!!
L621-624: This is wrong. Increase in blood glucose under stress is not due to "disturbed carbohydrate metabolism"!! It is a consequence of glycogen breakdown and gluconeogenesis. It is more under the control of cathecolemaines and cortisol, rather than insulin, glucagon ...
I see no discussion on cortisol.
The discussion section is disjointed. The authors discussed parameter by parameter, which is not acceptable.
Overall, other part of the discussion is not scientific, as the authors not measured many parameters before the heat stress.
Based of the rationale of the study design, deficient measurements, scientific issues in the discussion and poor presentation, I do not recommend this article for publication.
Comments on the Quality of English LanguageThere are many spelling issues in the test. Also, the text contains many non-formal sentences.
Author Response
Please see the attachment.

Reviewer 3 Report (Previous Reviewer 3)
Comments and Suggestions for Authors
The article Effects of Dietary Gamma-aminobutyric acid (GABA) Inclusion on Acute Temperature Stress Responses in Juvenile Olive Flounder (Paralichthys olivaceus) was received for evaluation. A species of fish that is important in aquaculture and breeds in waters between 20 and 25 degrees. Thus, its productivity will be affected due to climate change and the increase in seawater. The work is current and fundamental for aquaculture and for adaptations that the fish production sector needs due to the warming of ocean waters. Therefore, I classify its publication for the scientific community as a priority. The work is well written and has robust data, and After the revisions are carried out, I recommend approval. Congratulations to the authors for the excellent article.
Author Response
Please see the attachment.

Reviewer 4 Report (Previous Reviewer 2)
Comments and Suggestions for Authors
can be accepted
Round 2
Reviewer 1 Report (Previous Reviewer 4)
Comments and Suggestions for Authors
The authors continue to refer to the manuscript as if "supplementation" had been done in several parts of the text, as well as in the conclusion. This needs to be changed. Supplementation is something else; it is providing something more than already exists in the basal diet. Please review.
I agree with the other adjustments; it is very complete.
Easy to understand text. For me, ok.
But I'm not a native speaker of the language.
Author Response
Comment: The authors continue to refer to the manuscript as if “supplementation” had been done in several parts of the text, as well as in the conclusion. This needs to be changed. Supplementation is something else; it is providing something more than already exists in the basal diet. Please review. I agree with other adjustments; it is very complete.
Response: We sincerely appreciate the time you have taken to review our manuscript. Your insightful comments have helped us improve the clarity and quality of our work. We have carefully gone through the text and have changed “supplementation” to “inclusion” where needed as per the suggestion (the change is highlighted in yellow).
Once again, thank you for your support and constructive suggestions.
Reviewer 2 Report (Previous Reviewer 5)
Comments and Suggestions for Authors
I appreciate the authors effort to substantially revising the manuscript
Author Response
Comment: I appreciate the authors effort to substantially revising the manuscript.
Response: We sincerely appreciate your positive feedback and the time you have taken to review our manuscript. Your insightful comments helped us to significantly improve the quality of our manuscript to meet the standard of this journal.
This manuscript is a resubmission of an earlier submission. The following is a list of the peer review reports and author responses from that submission.
Round 1
Reviewer 1 Report
Comments and Suggestions for Authors
Excellent information and data. Accepted with changes. The wording should be revised. Be more precise and put data within the text to provide context for what is being mentioned. Increase and complement the discussion. Go deeper with the foundation and citations to the ideas raised. Both in method and discussion, there seems to be a dichotomy between the growth and stress part with blood parameters, metabolites, and gene expression. Review and make the sections more fluid and in the same style.
Abstract
Line 18, 28 – put the evaluated levels. Be more precise
Line 39 – average body composition data
Line 40 – average survival data
Line 43 to 47 – average data of the mentioned indicators
Line 52 to 54 – Be more precise in this statement, put more specifically the idea and conclusion of the study.
Introduction
Figure 1. Delete “This graph illustrates”; start at… the rate of change…
Line 88 – include data on what you mention as an acute deviation
Line 93 – time considered in secondary response
Line 95 – specify in time (h) prolonged exposure
Line 99 – delete word – broadly- or put more quotes where this idea is mentioned.
Line 110 – positively or negatively affected?
Line 113 – idem
Line 117 – state the objective directly. You can leave the research question, but state the objective of the research in a precise and specific way.
Line 127 – in Paralichthys olivaceus? Optimal for what?
Line 133 – Why is 70 ppm of GABA considered a control diet?
Table 1 – line 171 to 175 – was the determination done in triplicate? What is the n? 3, 4 or 5. Mention and if replications were performed, perform statistical analysis between diets by nutrient to emphasize that they are isoproteic, isolipidic and isoenergetic
Line 199 - Proximal composition of commercial diet
Line 200 - Progressive diet transition system, put percentage per day
Line 209 - What is the purpose of this previous test, put why
Line 215 - What are these optimums in water?
Line 266 - DO data
Results
Line 384 - Section, put data for the mentioned parameters
Table 3 - What is ns, put in figure caption.
Line 402 - What is n? Why use sem and not std?
Line 407 – data
Figure 3 – change or make more evident the differences in color of the different treatments in the graph to avoid confusion
Line 419 to 421 – move this sentence, it is part of the writing of the result, it should not be put in the caption of the figure - with a clear differentiation between groups 419 emerging after approximately 20 hours of exposure. Higher GABA concentrations, such as 420 GABA516, exhibited prolonged survival, while intermediate concentrations, such as GABA174 and 421 GABA396, showed the most declines in survival although, -
Line 424 – section – put general data
Line 429 – section – put precise data on each response variable (averages if there are no statistical differences)
Tables 4 and 5 – where there is no statistical difference put indicator in the name of the parameter and mention it in the caption of the table. Tables 4, if there are no statistical differences, remove the footnote where it mentions differences by row and put a note that there are no statistical differences. Why sem and not std?? Std is suggested
Table 6 – idem
Figure 4 and 5 – put in the figure caption, average and sem or std, n, meaning of vertical lines and note of no statistical significance
Discussion
Line 522 to 536 - put the growth and GABA supplementation values ​​that are being discussed to have the context of the differences and similarities. Put data and be precise
Line 537 – what do you mean by normal conditions? Temperature, GABA?
Line 538. Check the idea that is expressed carefully.
Line 538 to 547 - explain the idea well. Is GABA supplementation being taken to infer an improvement in growth, heat stress as a trigger or resistance to heat stress by GABA? Check carefully what is meant to be conveyed here
Line 549 – tilapia or olive flounder???
Line 550 to 552 – data for comparison
Line 553 to 555 – according to whom? Quotes on this assertion. Go deeper
Line 556 to 566 – go deeper into the observations and strengthen the discussion, there are no quotes or ideas to support or discuss beyond the authors’ observations.
Lines 5888 to 590 – important idea to develop further and support with more work and quotes
Line 591 – expand comparison and ideas based on this discussion.
Line 600 to 613 – good dissertation on results and possible effects and why, idea of ​​good adaptation strategies and environmental change effects. Strengthen with theoretical foundations and more citations to provide certainty and solid arguments to the authors' observations
Lines 616 to 621 – idem
Lines 622 to 628 – idem. Expand and strengthen by providing justifications for the discussion in the section.
Reviewer 2 Report
Comments and Suggestions for Authors
Animals
The study has potential, the introduction needs to be strengthened to provide a clearer context and research question. The methods and results sections require clarification and improvement in presentation. The discussion should be more comprehensive in synthesizing the findings and relating them to the broader literature. Additionally, the manuscript needs to be carefully edited to address minor errors in formatting and punctuation. With significant revisions, this manuscript could make a valuable contribution to the field. I recommend its major revision.
The introduction could be stronger in setting up the context and significance of the study. While it mentions the importance of GABA in stress mitigation, it would be helpful to provide more background on the specific challenges faced by olive flounder in aquaculture and how GABA supplementation could address these challenges.
The introduction could also benefit from a clearer statement of the research question and objectives.
The methods section is detailed, but some aspects could be clarified. For example, it would be helpful to know more about the specific GABA product used, including its purity and source.
Reduce the similarity to less than 15 % as presently its showing 26% that is higher side.
The section on statistical analysis could be more detailed, including information on the specific statistical software used and the criteria for determining significance.
Figure 4 and Figure 5 could be combined into a single figure with multiple panels to facilitate comparison between the different genes.
What do the changes in plasma metabolites and antioxidant enzymes suggest about the physiological response of olive flounder to temperature stress?
how do the results of this study compare to other studies on GABA supplementation in fish?
The discussion could also benefit from more critical evaluation of the study's limitations and potential avenues for future research.
The conclusion is brief and to the point, but could be more effective in summarizing the main findings and implications of the study.
There are some minor errors in formatting and punctuation throughout the manuscript. For example, some of the headings are not consistently formatted, and there are a few instances of missing or extra spaces.
Some of the sentences could be rephrased for clarity and concision. For example, the sentence "The specific mechanisms underlying the observed results warrant further investigation" could be rephrased as "Further research is needed to understand the mechanisms underlying these findings".
Comments on the Quality of English Languagea through revision is requested
Reviewer 3 Report
Comments and Suggestions for Authors
The article Investigating the Potential of Dietary Gamma-aminobutyric acid (GABA) Supplementation to Mitigate Acute Temperature Stress in Juvenile Olive Flounder (Paralichthys olivaceus). A species of fish that is important in aquaculture and breeds in waters between 20 and 25 degrees. Thus, its productivity will be affected due to climate change and the increase in seawater. The work is current and fundamental for aquaculture and for adaptations that the fish production sector needs due to the warming of ocean waters. Therefore, I classify its publication for the scientific community as a priority. The work is well written and has robust data, and I only indicate a few minor corrections.
I congratulate the authors for their relevant scientific contribution to the article's publication.

Reviewer 4 Report
Comments and Suggestions for Authors
The study by Ogun et al. aimed to investigate the potential of dietary gamma-aminobutyric acid (GABA) supplementation to mitigate acute temperature stress on the physiological resilience of juvenile olive flounder. GABA is one of the most researched additives in recent years, due to its diverse effects on the animal organism. However, it is not a common ingredient in diets, but rather used as an additive; this was also the case here. Therefore, the term supplementation used in the manuscript is not correct; replace it with "addition", "inclusion", among others. Supplementation means giving something that the animal is already eating in its diet orally in smaller doses; therefore, adjustments begin with the title and should be extended throughout the manuscript.
The text is very well written, with clear information and good results. However, there are still some points that need to be reviewed before publication:
a) in the abstracts section, different dosages were tested; but when presenting the results to the authors, they made a generalization. I suggest choosing the main results seen and being more specific in the description of these results. In the conclusion of this section, there was none; the authors preferred to talk about "implications"; remembering that the conclusion should respond to the objective.
b) I really liked your discussion, but it does not end with the clear objective of this research. add
c) in the production of the feed, with the different doses of GABA, it was not clear whether the authors made a pre-mix beforehand, since they are very small doses, this is important. The other processes were very well described; among them the data analysis, very detailed and clear.
d) The tables are very loaded with information, I suggest that the authors remove the main results of the research from these tables and present them in the form of a figure. Visually it is much more attractive; mainly because the vast majority of the variables analyzed did not differ statistically.
e) Why do you think there was a lack of results when fed with GABA? Is there any explanation? This research has some adverse factor that could explain this, since it was not expected. I think it is important to include a paragraph at the end of the discussion, making it clear to the authors, if any, the limitations of this research.
f) In figures 4 and 5, include the P value.
g) Since there is no clear defined objective, it is difficult to evaluate the conclusion, because it must respond to the objectives. I will wait for the adjustments before reevaluating this section.
Reviewer 5 Report
Comments and Suggestions for Authors
This study assessed the effects of dietary GABA supplementation on growth performance of olive flounder within an 8-week feeding trial and subsequent responses to acute heat stress. The authors mainly found that GABA had no benefits on fish production and heat stress responses. These results may be mainly due to that the authors not carefully considered previous data on this topic.
As mentioned in the discussion, but not in the introduction (!), GABA had no significant effects on growth performance of olive flounder under normal rearing conditions. Also, previous studies showed that GABA can be useful in growth promotion of this species under a chronic heat stress (28 days). Although this effect was observed when the fish fed a diet supplemented by 10000 mg/kg GABA (definitely higher than the doses used in the present study)! Taking these data in account together, the authors must avoid designing the study this way! On the other hand, chronic heat stress in more common in aquaculture facility than an acute one! Considering these drawbacks, I recommend not publishing this article.
Specific comments:
Introduction:
Fig. 1 is un-necessary.
Line 99-114: There are useful data regarding the use of dietary GABA in flounder, but the authors focused mainly on other species. I believe this has led to bias them in designing the study.
Methods:
The authors have only assessed the fish responses to the heat stress, but they did not determined the biochemical values before the stress. These data should be available to clarify "what happened in each group after the heat stress, compared to their NORMAL VALUES".
Discussion:
This section is a mixture of the repetition of the results and some general statements! It is not innovative and lacks clear connection among the different results.
Round 2
Reviewer 4 Report
Comments and Suggestions for Authors
The manuscript review was very well done. But the conclusion is not yet adequate; firstly because this section is to answer the objectives. Only this should be in the conclusion. reword it
The limitations section should be in the discussion, move this paragraph there
"A limitation of this study was the short duration of GABA inclusion, which may not have been enough to produce measurable effects on stress-related parameters. Future research should explore longer inclusion periods and varying stress durations to fully assess GABA's potential in stress mitigation. Furthermore, investigating the species-specific responses to GABA, along with factors like age, size, and genetic background, would provide a more comprehensive understanding of GABA’s applicability across different aquaculture settings."
Author Response
Reviewer 4’s Comments and Our Responses
Comment: The manuscript review was very well done. But the conclusion is not yet adequate; firstly because this section is to answer the objectives. Only this should be in the conclusion. reword it
The limitations section should be in the discussion, move this paragraph there
"A limitation of this study was the short duration of GABA inclusion, which may not have been enough to produce measurable effects on stress-related parameters. Future research should explore longer inclusion periods and varying stress durations to fully assess GABA's potential in stress mitigation. Furthermore, investigating the species-specific responses to GABA, along with factors like age, size, and genetic background, would provide a more comprehensive understanding of GABA’s applicability across different aquaculture settings."
Response: We have moved the paragraph to the discussion. Please see Lines 740-747
Reviewer 5 Report
Comments and Suggestions for Authors.
Author Response

(The authors gave the same response as above.)
